# Asynchronous release sites align with NMDA receptors in mouse hippocampal synapses

Shuo Li[1,2], Sumana Raychaudhuri[1], Stephen Alexander Lee[3,11], Marisa M. Brockmann [4], Jing Wang[5], Grant Kusick [1,6], Christine Prater[3,12,15], Sarah Syed[1,15], Hanieh Falahati[3,13,15], Raul Ramos [3,14,15], Tomas M. Bartol[7], Eric Hosy [8,9] & Shigeki Watanabe [1,10 ✉]

Neurotransmitter is released synchronously and asynchronously following an action potential. Our recent study indicates that the release sites of these two phases are segregated within an active zone, with asynchronous release sites enriched near the center in mouse hippocampal synapses. Here we demonstrate that synchronous and asynchronous release sites are aligned with AMPA receptor and NMDA receptor clusters, respectively. Computational simulations indicate that this spatial and temporal arrangement of release can lead to maximal membrane depolarization through AMPA receptors, alleviating the pore-blocking magnesium leading to greater activation of NMDA receptors. Together, these results suggest that release sites are likely organized to activate NMDA receptors efficiently.

[1] Department of Cell Biology, Johns Hopkins University, School of Medicine, Baltimore, MD, USA. [2] Department of Biological and Molecular Biology, Johns Hopkins Bloomberg School of Public Health, Baltimore, MD, USA. [3] Neurobiology Course, The Marine Biological Laboratory, Woods Hole, MA, USA. [4] Institute of Neurophysiology, Charité Universitätsmedizin, Berlin, Germany. [5] ThermoFisher Scientific, Hillsboro, OR, USA. [6] Biochemistry, Cellular and Molecular Biology Graduate Program, Johns Hopkins University, School of Medicine, Baltimore, MD, USA. [7] Salk Institute for Biological Studies, La Jolla, CA, USA. [8] Centre National de la Recherche Scientifique, Bordeaux, France. [9] Interdisciplinary Institute for Neuroscience, University of Bordeaux, Bordeaux, France. [10] Solomon H. Snyder Department of Neuroscience, Johns Hopkins University, School of Medicine, Baltimore, MD, USA. [11] Present address: Department of Biomedical Engineering, Columbia University, New York, NY, USA. [12] Present address: Department of Pharmacology and Neuroscience, School of Medicine, Texas Tech University Health Sciences Center, Lubbock, TX, USA. [13] Present address: Department of Neuroscience, Yale University School of Medicine, New Haven, CT, USA. [14] Present address: Department of Biology, Brandeis University, Waltham, MA, USA. [15] These authors contributed equally: Christine Prater, Sarah Syed, Hanieh Falahati, Raul Ramos. ✉email: shigeki.watanabe@jhmi.edu

Neurotransmitter is released synchronously within a millisecond of an action potential and asynchronously several milliseconds later[1]. Both phases of release result from exocytosis of synaptic vesicles at a specialized membrane domain: the active zone[2]. Within the active zone lies one or more release sites, individual units at which a single synaptic vesicle may fuse[3]. Our recent work suggests that release sites for synchronous and asynchronous release occupy unique domains within an active zone: synchronous release sites are uniformly distributed, while asynchronous release sites are abundant near the center of an active zone[4]. However, the functional importance of this spatial organization is unknown.

For excitatory signaling in the central nervous system, glutamate released from presynaptic boutons activates receptors on the postsynaptic membrane. Two ionotropic receptors are critical: α-amino-3-hydroxy-5-methyl-4-isoxazolepropionic acid (AMPA) receptors and N-methyl-D-aspartate (NMDA) receptors[5,6]. These receptors are recruited to the receptive field by scaffolding proteins that make up electron-dense cytomatrix, or postsynaptic density (PSD)[7]. Given the low sensitivity of these receptors for glutamate binding[8–11], activating them requires high concentrations of glutamate. Thus, the spatial organization of synchronous and asynchronous release sites may be important for the activation of receptors by increasing the glutamate concentration near the receptors.

A large body of work suggests that the number of these receptors, rather than their location relative to release sites, is the main determinant of how signals are transmitted across synapses. At mammalian central synapses, the number of AMPA and NMDA receptors is highly variable, ranging from zero to several hundreds[12–18]. However, the number scales with the size of postsynaptic densities[16–20], and thus, the postsynaptic membrane is highly occupied by the receptors. Given ~2000 glutamate molecules released per fusion event[21], one would expect these receptors to be activated efficiently regardless of where in the PSD they are located relative to where glutamate is released. In fact, some computer simulations suggest that the receptor number is more important for the amplitude of synaptic signaling than the locations of release[18,22,23].

In contrast to this view, recent studies suggest that receptor activation is greatly influenced by receptors forming clusters that are positioned close to release sites[24]. Due to rapid diffusion[25–27], the concentration of glutamate necessary for maximal AMPA receptor activation is only achieved at the point of release. In fact, several computer simulations demonstrate that the open probability of AMPA receptors is reduced by 20–70% if located 100 nm away from the point of release[18,25,28,29]. Several electrophysiology experiments also suggest that glutamate release from single vesicles does not saturate postsynaptic receptors[30–32]. In addition, localization studies suggest that AMPA receptors are clustered within the PSD[33,34] and segregated from the NMDA receptor clusters[35]. Release sites are aligned with clusters of AMPA receptors[36], and their association through transsynaptic adhesion proteins affects the magnitude of synaptic transmission[37]. Thus, where glutamate is released relative to receptors may be important for their activation.

The timing of glutamate release is also a key component for receptor activation, particularly for NMDA receptors. NMDA receptors must bind two glutamate molecules to activate[38,39]. Depending on the concentration of glutamate, only a single binding site may be occupied, leading to desensitization rather than activation[38,39]. However, the single-bound state can last for tens of milliseconds, during which the second release can favor their activation[38,39]. In addition, at resting membrane potential the pore of NMDA receptors is blocked by magnesium ions, which must be removed by membrane depolarization[40–42].

Classically, paired stimuli are applied for NMDA receptor activation[43–45]. However, a single stimulus can potentially prime NMDA receptors for activation, with synchronous release depolarizing the postsynaptic membrane and asynchronous release providing extra glutamate. Thus, both the location and timing of glutamate release are likely important for determining how signals are transmitted at excitatory synapses.

To test whether asynchronous release sites have a spatial relationship with receptors, we developed an approach to localize fusion pits and receptors at the ultrastructural level. We demonstrate that synchronous and asynchronous release sites are aligned with clusters of AMPA and NMDA receptors, respectively. Computer simulations suggest that this organization can induce membrane depolarization through the AMPA receptors and activate NMDA receptors more efficiently. These data indicate that one potential role of this spatial organization of synchronous and asynchronous release sites is to prime NMDA receptors for activation.

## Results

**Validation of small-metal affinity staining of His-tag proteins**. To reveal the spatial and temporal relationship between release sites and receptors, we need an approach to visualize receptors relative to fusions events observed by electron microscopy. To this end, we developed a method to label these receptors with gold particles using a high-affinity interaction between nickel and polyhistidine (His-tag, hereafter; Fig. 1a)[46]. GluA2 (AMPA receptor subunit) tagged on its extracellular domain with His-tag or HaloTag[47] was expressed in the cultured wild-type mouse hippocampal neurons using lentivirus. HaloTag::GluA2 served as a negative control to test the specificity of Ni-NTA-gold labeling. We incubated these neurons with Ni-NTA-gold (5 nM) for 30 min and subjected them to high-pressure freezing. Frozen samples were then prepared for electron microscopy, and 40-nm-thick sections collected. About 100 electron micrographs were acquired per experiment from each condition and quantified blinded to treatment (Supplementary Fig. 1a, e.g., micrographs), and each experiment was repeated three times. In the wild-type neurons expressing His-tag::GluA2, ~70% of synaptic profiles contained gold particles in the synaptic cleft; the median number of gold particles was 2.6 per synaptic profile (Supplementary Fig. 1c, d; note that synaptic profiles are random 40-nm segments of synapses). The amount of staining is consistent with antibody-based approaches[12–15]. In contrast, almost no gold particles were observed in controls expressing HaloTag::GluA2 (~4% synaptic profiles, 0.03 gold/synapse, Supplementary Fig. 1a, c, d and Supplementary Table 1 for details), suggesting that the labeling is specific.

We next repeated the same experiments in GluA2 knock-out neurons ($Gria2^{-/-}$) to test if the overexpression of the GluA2 in wild type would significantly change the number of receptors at the PSD. No differences were observed between wild-type and knock-out neurons (Supplementary Fig. 1b–f). These results suggest that this affinity-based labeling can report the distribution of receptors at the ultrastructural level. We named this approach small-metal affinity staining of His-tag, or SMASH.

**AMPA and NMDA receptors are segregated within the PSD**. With the labeling approach validated, we next measured the locations of AMPA and NMDA receptors within the PSD. We expressed either His-tag::GluA2 or His-tag::NR1 in wild-type neurons. In single profiles, AMPA receptors were biased toward the edge of the PSD ($p < 0.001$), whereas NMDA receptors were uniformly distributed (Fig. 1b–d; $p > 0.8$), suggesting they may occupy different domains within the PSD (Fig. 1b–d; $p < 0.001$).

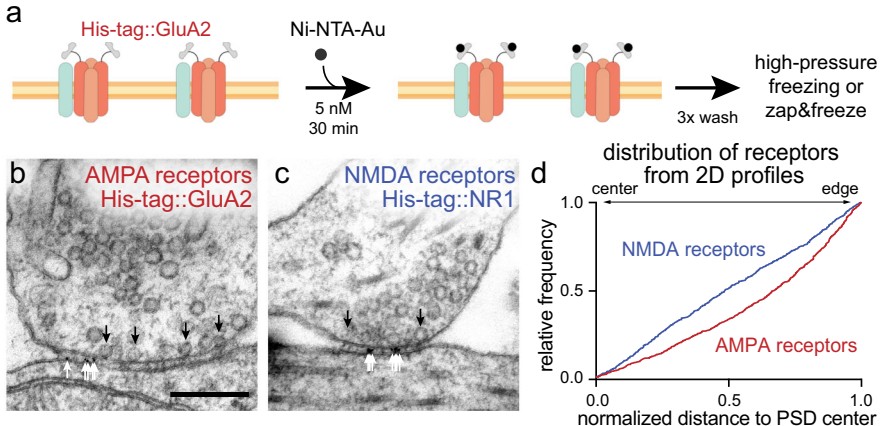

**Fig. 1 AMPA and NMDA receptors cluster at the periphery and the center of the postsynaptic density, respectively. a** Schematic of the small-metal affinity staining of His-tag (SMASH) strategy for live-cell labeling of overexpressed His-tagged surface GluA2 (AMPA receptors) with nickel-NTA-gold (5 nm). Example transmission electron micrographs of synapses after SMASH labeling and high-pressure freezing, showing gold particles in the synaptic cleft of wild-type neurons expressing His-tag::GluA2 (**b**) and His-tag::NR1 (**c**). Black arrows: docked synaptic vesicles. White arrows: gold particles. **d** Cumulative relative frequency distribution of lateral distances from gold particles to the center of the postsynaptic density (PSD) from 2D profiles. Distances are normalized to the length of the PSD: a gold particle at 0 would be exactly at the center and at 1 exactly at the edge. AMPA receptors are biased toward the edge (median = 0.7, $p < 0.001$, $n = 891$ particles, $N = 4$ cultures), while NMDA receptors are uniformly distributed (median = 0.5, $p > 0.8$, $n = 746$ particles, $N = 4$ cultures). Bias of particle locations toward the center or edge of the postsynaptic density in (**d**) was tested by comparing each group to a theoretical median of 0.5 using one-sample two-tailed Wilcoxon signed-rank tests. See Supplementary Data Table 1 for full pairwise comparisons and summary statistics.

However, single synaptic profiles are random representations of synapses; each image represents a 40-nm segment of a synapse, sliced at a random location. These slices are not necessarily going through the center of an active zone/PSD. Nonetheless, the data from synaptic profiles can be used to estimate the distributions when sufficient data are collected (Supplementary Fig. 2a). To more accurately localize receptors, we reconstructed synapses using spin-mill serial block face imaging (Fig. 2a, Supplementary Fig. 2, and Supplementary Movies 1–4, $N = 2$; see "Methods" for details). Unlike traditional serial block face imaging approaches, spin milling using an oxygen plasma ion beam enables imaging of large areas ($400 \times 400$ μm) with isotropic resolution of ~5 nm[48]. At least 17 synapses were reconstructed from each sample, and experiments repeated from two independent samples. Similar to single profiles, AMPA and NMDA receptors were differentially distributed in the reconstructed postsynaptic densities (Fig. 2b, $p < 0.001$). As in single profiles (Fig. 1d), AMPA receptors were biased toward the edge (Fig. 2b, $p < 0.001$, and Supplementary Fig. 2b for more examples). Interestingly, NMDA receptors were enriched around the center of the PSD (Fig. 2a, b, $p < 0.001$, and Supplementary Fig. 2d for more examples). The distributions of these receptors are consistent with previous experiments using single-molecule localization microscopy[33,35] and electron microscopy[14,49]. These data suggest that AMPA and NMDA receptors are differentially localized within the PSD.

**AMPA and NMDA receptors are clustered in the PSD**. Both AMPA and NMDA receptors qualitatively seemed clustered[33] in the reconstructed synapses. To test this quantitatively, we performed K-means cluster analysis on the reconstructed synapses (Fig. 2c, d and Supplementary Fig. 2c, e). The same analysis was then repeated 50 times with particle locations randomized within the PSD for each synapse. The sums of square differences were compared between the actual locations and randomized locations (see "Methods" for detail). The areas of postsynaptic densities were similar between samples (Fig. 2e), and thus the data are not normalized. The median numbers of the AMPA receptor and NMDA receptor clusters were 2 and 1 per synapse, respectively,

based on the K-means analysis (Fig. 2f). The median numbers of AMPA and NMDA receptors were 8 and 6 per cluster (Fig. 2g, ranges: 4–23 AMPA receptors and 4–18 NMDA receptors), or 16 and 10 per synapse, respectively (Fig. 2h, full ranges: 0–58 AMPA receptors and 0–55 NMDA receptors). These numbers correspond to 200 and 135 per μm², respectively (full ranges: 0–1021 AMPA receptors and 0–1079 NMDA receptors) and comparable to previous estimates from the freeze-fracture immuno-gold labeling of adult rat cerebellum[17] and immuno-electron microscopy of rat hippocampus[19]. Since ~20 AMPA receptors are likely available in each nano-domain (ranges: 7.3–42.2)[33], our labeling efficiency is likely ~50 %. Nonetheless, these results are consistent with the single-molecule localization study[35] and suggest that AMPA receptors tend to form ~2 clusters (Fig. 2f) and surround the NMDA receptor cluster, located near the center of the PSD.

**Asynchronous release sites are aligned with NMDA receptors**. Recent studies suggest that release sites are trans-synaptically aligned with AMPA receptors[36]. Our data indicate that NMDA receptors occupy different domains within the PSD (Figs. 1d and 2b). Thus, it is not clear if NMDA receptors also align with release sites. Interestingly, their distribution near the center of the PSD mirrors the recently described distribution of asynchronous release sites in the active zone[4]. To test whether asynchronous release sites and NMDA receptors are aligned, we performed zap-and-freeze experiments after SMASH labeling (Fig. 3a, b). Neurons expressing SnapTag::GluA2 or SnapTag::NR1 were used as controls for background gold staining (Supplementary Fig. 3a–f). Specifically, we stimulated neurons expressing fusion proteins once and froze them after 5 and 11 ms. Our recent study suggested that fusion intermediates captured at 5 ms represent remnants of synchronously fusing vesicles, while those at 11 ms represent asynchronously fusing vesicles, since the treatment with EGTA-AM only eliminated the latter events[4]. The numbers and distributions of docked vesicles and exocytic pits were all consistent with our previous studies (docked: $1.9 \pm 0.05$ per synaptic profile; pits: $0.28 \pm 0.03$ per synaptic profile Fig. 3c,

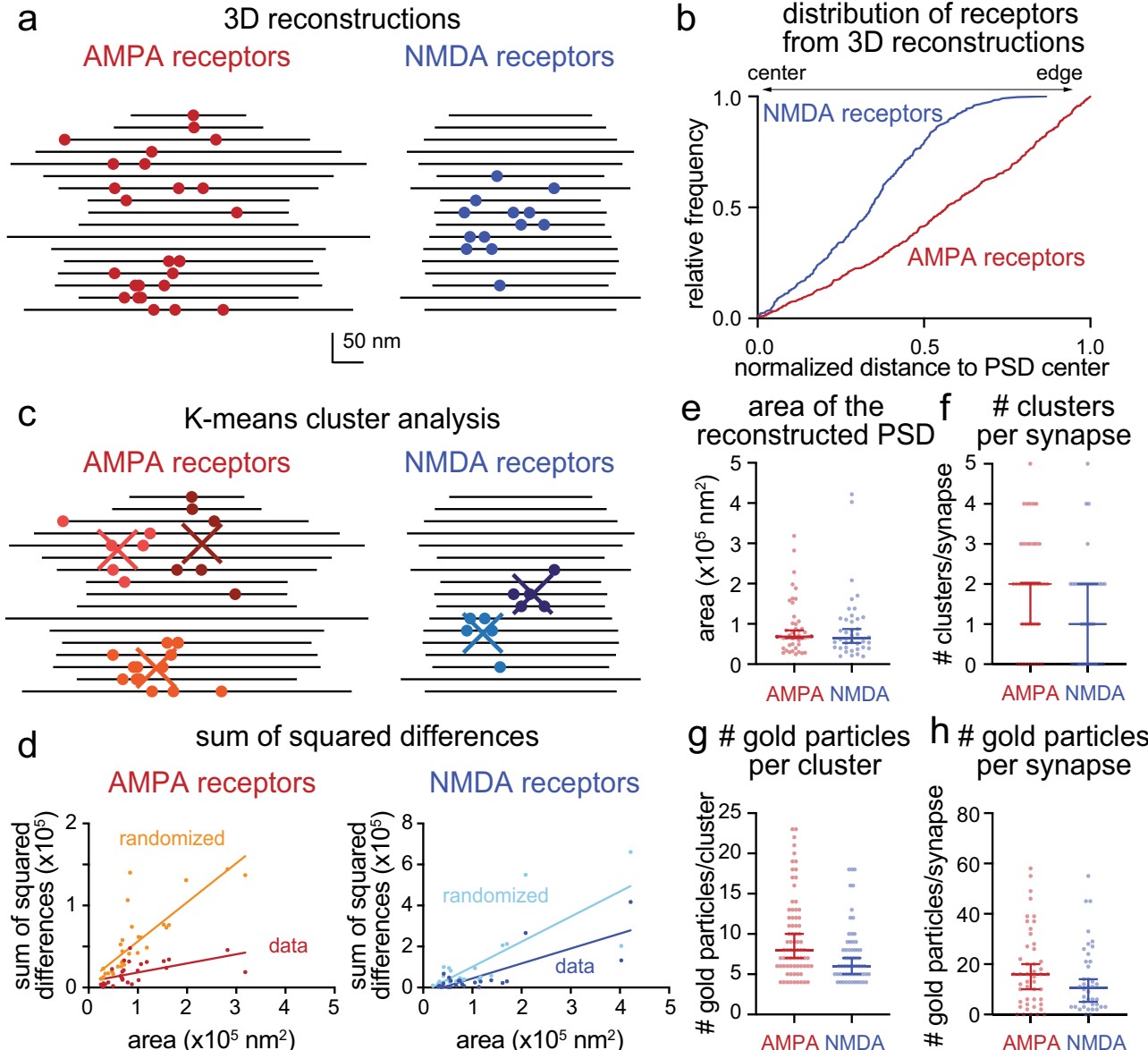

**Fig. 2 AMPA and NMDA receptors cluster at the periphery and the center of the reconstructed postsynaptic density, respectively. a** Examples of synapses from spin-mill serial block face scanning electron microscopy; each line indicates the extent of the cleft in a single 20-nm-thick 2D profile, each circle indicates the location of a gold particle. **b** Cumulative relative frequency distribution of lateral distances from gold particles to the center of the postsynaptic density (PSD) from 3D profiles. Distances are normalized to the size of the PSD and corrected as fractional area assuming a circular PSD: a gold particle at 0 would be exactly at the center, at 1 exactly at the edge, and at 0.25 equidistant between center and edge. AMPA receptors are slightly biased toward the edge (median = 0.6, $p < 0.001$, $n = 840$ particles, $N = 2$ cultures), while NMDA receptors are biased toward the center (median = 0.3, $p < 0.001$, $n = 550$ particles, $N = 2$ cultures). **c** Same as a, except showing the centers of clusters determined by k-means clustering. **d** Sum of squared differences, calculated from each particle to the centroid of the cluster. Data: the actual distances of gold particles to their cluster center. Randomized: the distances of gold particles to their putative cluster center after randomizing the locations of gold particles at each synapse. Each dot: a synapse. Simple linear regression test: AMPA receptors, data, $R^2 = 0.32$, randomized = 0.67, $p < 0.001$; NMDA receptors, data, $R^2 = 0.63$, randomized = 0.62, $p = 0.04$. **e** Areas of the reconstructed postsynaptic densities. Each dot: a single reconstructed synapse. Error bars: median and 95% confidence interval, $p > 0.6$, Mann–Whitney test. **f** Number of clusters per synapse determined by k-means clustering. Each dot: a single reconstructed synapse. Error bars: median and 95% confidence interval, $p = 0.06$, Mann–Whitney test. Number of gold particles per cluster (**g**) and per synapse (**h**). Bias of particle locations toward the center or edge of the postsynaptic density in (**b**) was tested by comparing each group to a theoretical median of 0.5 using one-sample two-tailed Wilcoxon signed-rank tests. See Supplementary Data Table 1 for full pairwise comparisons and summary statistics.

Supplementary Fig. 3g, h)[4,50]. Of note, asynchronous fusion intermediates at 11 ms were strongly biased toward the center (Fig. 3a, b, d, median = 0.1, $p < 0.001$, and Supplementary Fig. 3a–d for more example micrographs). Thus, the distribution of fusion events during asynchronous release is similar to that of NMDA receptors.

To test the spatial relationship between fusion events and receptors, we measured the distance between receptors and docked vesicles or exocytic pits. The median distances from docked vesicles to AMPA and NMDA receptors were 95 and 73 nm, respectively, at rest (Fig. 3e, inset), and remained largely unchanged following stimulation (Fig. 3e, inset: AMPA receptors,

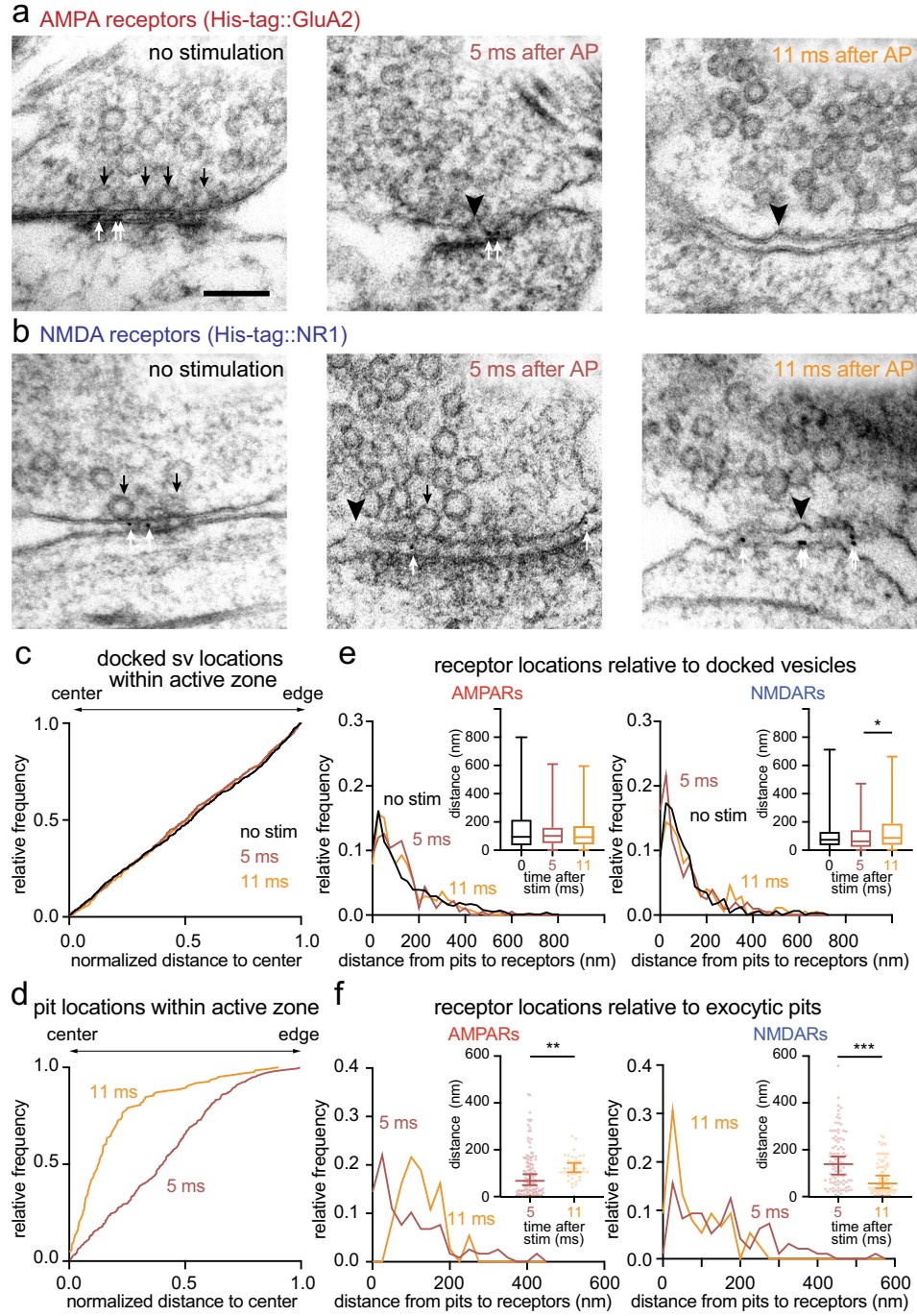

**Fig. 3 Synchronous and asynchronous release are aligned to AMPA and NMDA receptors, respectively.** Example transmission electron micrographs of synapses after SMASH labeling and high-pressure freezing at the indicated time points after an action potential (AP), showing pits (black arrowheads), docked vesicles (black arrows), and gold particles (white arrows) at synapses of wild-type neurons expressing His-tag::GluA2 (**a**) and His-tag::NR1 (**b**). Cumulative relative frequency distribution of lateral distances between docked vesicles (**c**) or exocytic pits (**d**) and the active zone center at 5 or 11 ms after stimulation, as measured within 2D profiles by transmission electron microscopy. Exocytic pits were biased toward the center both at 5 ms (median = 0.4, $p < 0.001$, $n = 286$ pits, $N = 7$ samples) and 11 ms (median = 0.13, $p < 0.001$, $n = 124$ pits, $N = 7$ samples), but pits at 11 ms were strongly biased toward the center ($p < 0.001$, Kolmogorov–Smirnov $D$ test). Bias of vesicles and pits locations toward the center or edge of the active zone in (**c**, **d**) was tested by comparing each group to a theoretical median of 0.5 using one-sample two-tailed Wilcoxon signed-rank tests. Relative frequency distribution of lateral distances between docked vesicles (**e**) or exocytic pits (**f**) and gold particles within the cleft at 5 ms (red) and 11 ms (orange) after stimulation. Insets: the same data plotted as box and whiskers for docked vesicles (**e**) and scattered dots for pits (**f**) to show the median distances. Error bars: median and 95% confidence interval for the scattered dot plots. Kolmogorov–Smirnov $D$ tests were performed in each case, with additional post hoc Dunn's multiple comparisons tests for docked vesicle data. *$p < 0.05$, **$p < 0.01$, ***$p < 0.001$. Experiments were repeated three times independently. Scale bar: 100 nm. See Supplementary Data Table 1 for full pairwise comparisons and summary statistics.

102 nm at 5 ms; and 92 nm at 11 ms; Fig. 3e, inset: NMDA receptors, 63 nm at 5 ms, 88 nm at 11 ms). This relationship between docked vesicles and receptors is expected given the uniform distribution of docked vesicles in the active zone before and after stimulation (Fig. 3c)[4].

In contrast, the distribution of exocytic pits relative to receptors was not uniform. During the synchronous phase of release (5 ms time points in our assay), pits were distributed throughout the active zone (Fig. 3d). However, when measured relative to each type of receptors, exocytic pits were found closer to AMPA receptors (median = 67 nm) than NMDA receptors (median = 139 nm; Fig. 3f). Interestingly, exocytic pits during asynchronous phase of release (11 ms) were distant from AMPA receptors (Fig. 3f, median = 120 nm) but closer to NMDA receptors (Fig. 3f, median = 56 nm). These results suggest that neurotransmitter is likely released synchronously near AMPA receptors and asynchronously around NMDA receptors.

**The location, but not the sequence, of release is important for NMDA receptor activation.** The spatial organization of release sites and receptors and temporal sequence of their usage could allow preferential activation of NMDA receptors. First, synchronous release activates AMPA receptors, depolarizing the postsynaptic membrane and alleviating $Mg^{2+}$ block of NMDA receptors[40,41]. Asynchronous release then increases the glutamate concentration in the synaptic cleft to favor the activation of NMDA receptors, potentially of those in the single-bound state. To test this possibility, we need to assess how NMDA receptors respond to asynchronous release in the absence of a $Mg^{2+}$ block and compare it to the response following AMPA receptor-mediated membrane depolarization in the presence of $Mg^{2+}$ block. In addition, the locations of synchronous and asynchronous release must be swapped to assess the importance of the spatial organization. To address these points, we performed computational simulations using the MCell platform that we have recently developed[27,37]. This model simulates receptor activation in dendritic spines by incorporating realistic synapse morphology as well as the measured kinetics of molecules[27,37]. We further incorporated the observed distributions of receptors and their numbers revealed by super-resolution imaging[35] (Fig. 2): clusters containing ~20 AMPA receptors and ~15 NMDA receptors were placed around the periphery and center of the PSD, respectively. The centroid to centroid distance between the clusters was set at 100 nm[37]. We simulated the activation of AMPA and NMDA receptors with a sequence of two release events, with one occurring near the AMPA receptor cluster and another occurring near the NMDA receptor cluster. The timing of these two release events was varied from 0 to 50 ms apart (0 ms means simultaneous release at these two locations).

Using this model, we first determined how AMPA and NMDA receptors behave in response to asynchronous release in the absence of $Mg^{2+}$. Asynchronous release increased the responses of AMPA receptors by 50% and NMDA receptors by 84%, when compared to the responses from a single release (Supplementary Fig. 4a–f). For AMPA receptors, the desensitization of the receptors[51,52] hampered their response to asynchronous release during the first 15 ms (Supplementary Fig. 4a–c). In contrast, the responses of NMDA receptors were higher as the timing of the release was delayed, reaching a 122% increase at 50 ms (Supplementary Fig. 4d–f). This increase is likely due to the binding of glutamate to those single-bound receptors since this state can be maintained for tens of milliseconds and is favored for activation by the second release. These results suggest that asynchronous release favors the activation of NMDA receptors in the absence of a $Mg^{2+}$ block. Interestingly, the proportion of

NMDA and AMPA receptors activated by these two release events was similar to the proportion activated when a single vesicle release occurs near NMDA receptors, but not around AMPA receptors or at random locations (Supplementary Fig. 4g). These results suggest that the location of release also influences the activation of NMDA receptors.

We next tested the effect of the AMPA receptor-mediated membrane depolarization on NMDA receptor activation in the presence of $Mg^{2+}$. Two release events were induced simultaneously, while the membrane potential was depolarized by 30 or 45 mV, mimicking the changes in membrane potential after a single release event near AMPA receptors or after simultaneous release at both AMPA and NMDA receptors, respectively (Fig. 4a). The constant voltage at resting potential was used as a control. 45 mV depolarization doubled NMDA receptor activation compared to no membrane potential change (Fig. 4a, b). This response is substantially higher than the responses from a single release near the AMPA receptors (~4× increase) or NMDA receptors (~3× increase; Fig. 4a, b). These results suggest that NMDA receptors can be efficiently activated by two release events (multivesicular release)[53,54], which are prevalent in these neurons[4].

To test whether the timing of asynchronous release is important for NMDA receptor activation, we varied the timing of the second release by 5, 8, 10, and 20 ms (Fig. 4c). The kinetics of depolarization and repolarization following AMPA receptor activation are integrated into the platform[27,37,55]; the depolarization peaks between 3 and 5 ms after the release, increasing by ~25 mV, and declines to 2/3 of the maximum after 8 ms and almost to the resting potential by 10 ms (Supplementary Fig. 4h). The response from NMDA receptors peaked when asynchronous release occurred at 5 ms and decreased progressively as the asynchronous release was further delayed (Fig. 4d). These results are in sharp contrast to the activation of NMDA receptors in the absence of $Mg^{2+}$, indicating that the kinetics of membrane depolarization and repolarization, and thereby $Mg^{2+}$ unbinding and binding, determine the activity of NMDA receptors.

To test the importance of the location and order of release, we either flipped the order of release (first on NMDA receptors, and then on AMPA receptors) or applied both release at the same locations (Fig. 4e). The two release events were paired 5 ms apart to test the maximal response. Flipping the order did not change the response of NMDA receptors (Fig. 4f). However, when both release events were applied near NMDA receptors consecutively, the response was 19% lower (Fig. 4f), presumably because the membranes cannot be maximally depolarized. In fact, when we simulated with a higher level of depolarization, matching the degree of depolarization expected from releasing near AMPA receptors, the response of NMDA receptors was much stronger (~32% increase, Fig. 4f and Supplementary Fig. 4i, j), suggesting activation of AMPA receptors is essential for NMDA receptor activity. In fact, two consecutive release events near AMPA receptors leads to a better activation of NMDA receptors (Fig. 4f), but this increase occurs at the expense of an increased number of desensitized AMPA receptors and thereby faster synaptic depression. Together, these results suggest that the transsynaptic alignment of release sites and receptors likely ensures the maximal depolarization through AMPA receptors and thereby efficient activation of NMDA receptors, while avoiding saturation of AMPA receptors from a single stimulus (Fig. 5).

## Discussion

Here we demonstrate the transsynaptic alignment of synchronous and asynchronous release sites with AMPA receptor and NMDA receptor clusters, respectively. These findings have implications

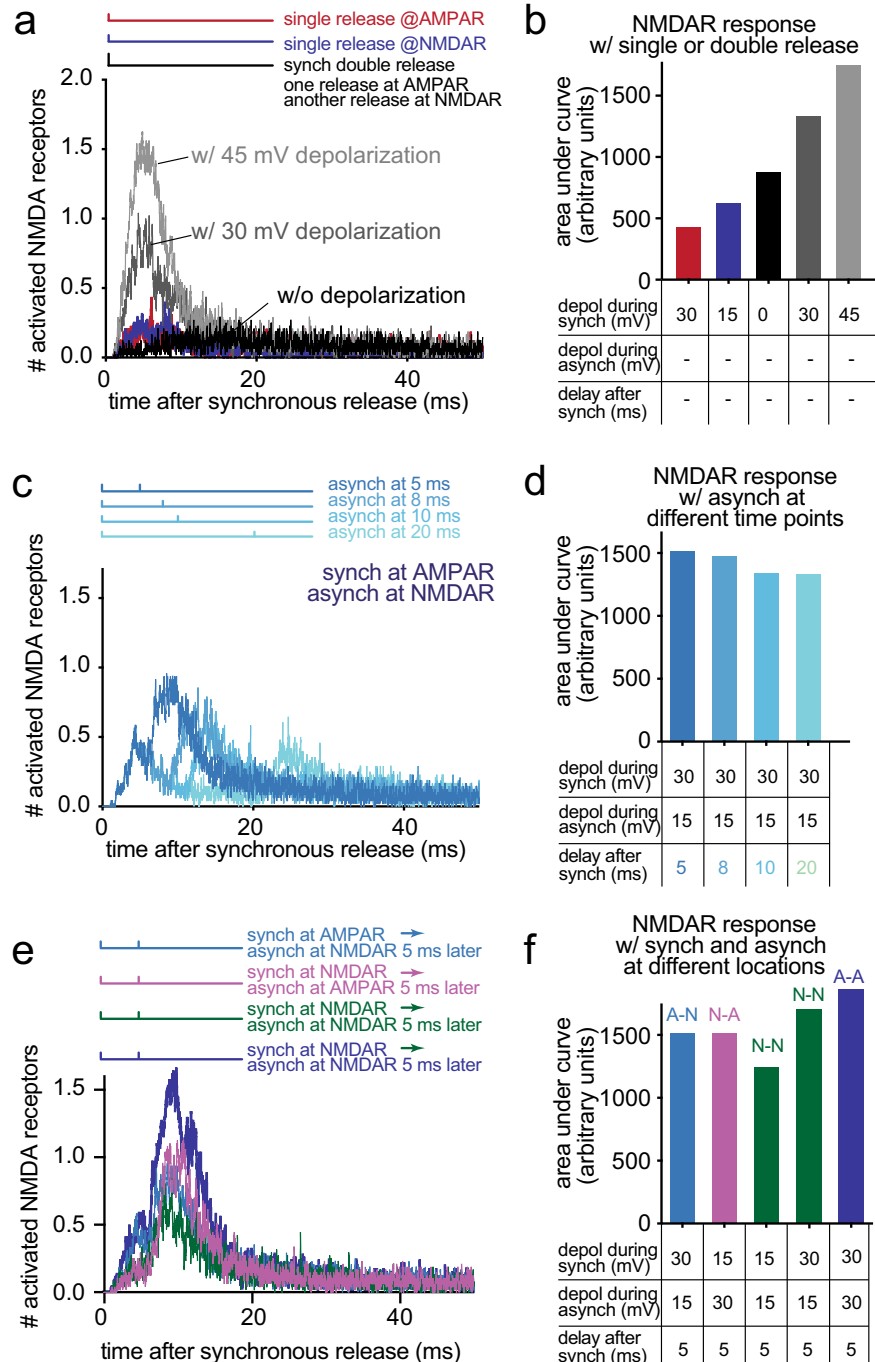

**Fig. 4 Computer simulations predict better activation of NMDA receptors with asynchronous release. a** Time course of simulated NMDA receptor activation, resulting from one or two release events at the indicated locations while varying the degree of depolarization of the postsynaptic membrane. The number averaged from 48 simulations is plotted. The vertical lines in the diagrams shown above the plot indicate when synchronous and asynchronous release occur. Synchronous release always occurs at time 0. Double release means two release events happen simultaneously. Synch synchronous, Asynch asynchronous. **b** The area under curve calculated from each data set in (**a**) and plotted as a bar graph. The locations of release are described in (**a**), and the degree of depolarization (depol) and the delay between synchronous and asynchronous release used in simulations are listed at the bottom. **c** Same as (**a**), but varying the interval between two release events. **d** Same as (**b**), but plotted from each data set in (**c**). **e** Same as in (**a**), but varying the order of the release. **f** Same as in (**b**), but plotted from each data set in (**e**) and plotted as a bar graph. A-N: synchronous release at AMPA receptors and asynchronous release at NMDA receptors. N-A: synchronous release at NMDA receptors and asynchronous release at AMPA receptors. N-N: both synchronous and asynchronous release at NMDA receptors. A-A: both synchronous and asynchronous release at AMPA receptors.

for how signals are transmitted at synapses and how release sites and receptors are organized.

The presence of the receptor clusters and their transsynaptic alignment with release sites suggests that where glutamate is released relative to receptors is likely important for their activation. This is in sharp contrast with the original view that large numbers of receptors are present in the postsynaptic receptive field[51] with no particular pattern of localization, and that

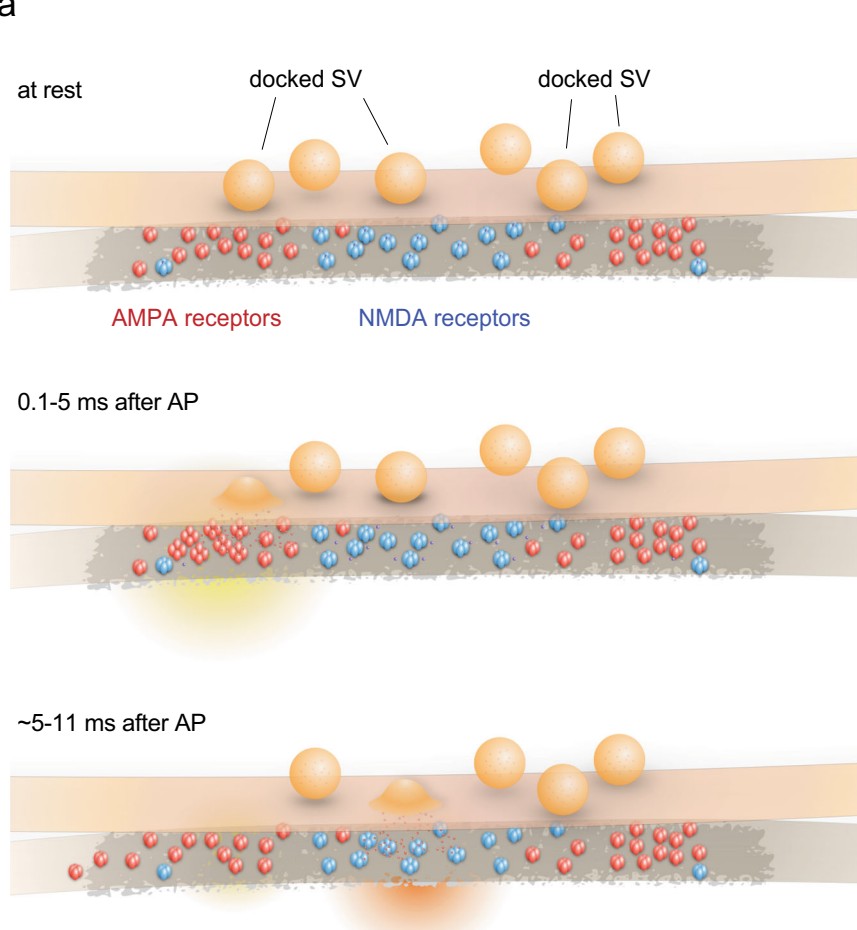

**Fig. 5 Proposed synaptic organization and events. a** Schematic of proposed synaptic organization and events. Docked vesicles are found throughout the active zone. AMPA receptors are found toward the edge, while NMDA receptors are biased toward the center. Synchronous fusion begins within hundreds of microseconds near the AMPA receptor cluster. Released glutamate activates AMPA receptors, which, in turn, depolarizes the membrane and alleviates the $Mg^{2+}$ block of NMDA receptors. Between 5 and 11 ms, residual calcium triggers asynchronous fusion, preferentially toward the center of the active zone and across from the NMDA receptor cluster, favoring the NMDA receptors. Although shown here as taking place in the same active zone, the degree to which synchronous and asynchronous release may occur at the same active zone after a single action potential is unknown. This transsynaptic organization allows the maximal depolarization of the postsynaptic membrane and efficient activation of NMDA receptors.

glutamate released from single vesicles anywhere in the active zone can efficiently activate both AMPA and NMDA receptors[5,56]. However, several lines of recent data seem to support the idea that release in the proximity of receptors is a major determinant of synaptic strength. First, release evoked by an action potential does not saturate receptor activation[31]. Second, activation of AMPA receptors is sharply dependent on their distances from the point of release[18,25,28,29]. Third, AMPA receptors need to be in the clusters to enhance the amplitude of signals. Simply increasing the number of AMPA receptors in postsynaptic densities using optogenetics does not change the quantal amplitude nor strength of response from existing synapses, suggesting that the location of receptors is more important[57]. Fourth, the alignment is modulable and can potentially alter synaptic strength[36,58]. Our data are consistent with these studies and further support the idea that the precise location of release influences efficiency of receptor activation at excitatory synapses.

In addition, our results demonstrate that the NMDA receptor activation is also influenced by such transsynaptic nano-architectures. NMDA receptors have a higher affinity for glutamate, and thus their locations within a PSD are thought to be less critical. Previous simulations demonstrated that whether they are in the cluster or randomly localized, NMDA receptors can be activated equally well from a single vesicle release[35]. Our data here also suggest that locations of release are less important for NMDA receptor activation as long as AMPA receptors are activated. In fact, NMDA receptors are activated to a greater degree when both synchronous and asynchronous release occur near AMPA receptors. However, this release pattern would increase the number of desensitized AMPA receptors, leading to faster depression at these synapses. Thus, one release event near AMPA receptors and another release event near NMDA receptors likely maximize the membrane depolarization and NMDA receptor activation, while ensuring that a sufficient number of naive AMPA receptors are available to respond to the next stimulus.

Whether glutamate is released spontaneously from single vesicles or actively following an action potential, similar proportions of AMPA and NMDA receptors are thought to be activated[5]. Our data indicate that an action potential may induce glutamate release from two vesicles in some synapses:

synchronously near AMPA receptors and asynchronously near NMDA receptors. Although we do not know how often a synapse releases glutamate both synchronously and asynchronously after a single action potential, evoked release likely leads to greater activation of NMDA receptors, given that synchronous multivesicular release is also quite prominent in these synapses[4]. Therefore, to achieve a similar proportion of activation by single vesicles, glutamate may need to be released at a particular location during spontaneous release. Our computer simulations suggest that a single vesicle release near the NMDA receptors may be able to activate both AMPA and NMDA receptors with a similar proportion to the evoked release, suggesting that spontaneous release may occur near NMDA receptors. However, spontaneous release has been proposed to use a distinct pool of synaptic vesicles[59–61], which may not be readily available at active zones. In addition, a distinct set of postsynaptic receptors may be activated by spontaneous release[62,63]. In fact, the NMDA receptor activated during spontaneous release does not seem to depend on the dendritic depolarization or AMPA receptor activity[64]. Thus, spontaneous release may elicit the postsynaptic currents by a completely different mechanism. However, these ideas are still contentious and require further testing[65].

Here, we propose that transsynaptic organization of NMDA receptors may allow them to be activated during asynchronous release (Fig. 5). However, it is possible that this organization may serve different functions. To test this point, the actual functional output from NMDA receptors must be measured experimentally while manipulating their organization. Currently, it is unknown how asynchronous release sites are aligned with NMDA receptors. One possibility is that this organization arises by coincidence, that is, asynchronous release sites simply occur further away from synchronous release sites, which may be aligned with AMPA receptor clusters via neuroligin-1[37], and coincide with NMDA receptors. Alternatively, many synaptic adhesion molecules exist, and they all interact with the presynaptic release machinery as well as postsynaptic receptors and their scaffolding proteins[24,37,66,67]. Thus, it is tempting to speculate that the arrangement of these molecules gives rise to this unique transsynaptic organization of release sites and receptors at excitatory synapses.

## Methods
All animal procedures were performed according to the National Institute of Health guidelines and were approved by the Animal care and Use Committee at the Johns Hopkins University School of Medicine.

**Animals**. Wild-type mice (C57/BL6J) were purchased from Charles River Laboratory and Gria2$^{(+/−)}$ mouse line (*Gria2$^{tm1Rlh}$*/J) was kindly provided by Dr. Richard Huganir. Gria2$^{(+/−)}$ line was maintained as heterozygous and knock-out Gria2$^{(−/−)}$ pups at postnatal day 0 (P0) were used for neuronal culture.

**Neuron culture**. Both astrocytes and hippocampal neuron cultures were established from embryonic day 18 or P0 wild-type animals. Both sexes were indistinguishably used in this study. Astrocytes were harvested from cortices with trypsin treatment for 20 min at 37 °C with shaking, followed by dissociation and seeding on T-75 flask. Astrocytes were grown in DMEM supplemented with 10% FBS and 0.1% penicillin-streptomycin at 37 °C and 5% $CO_2$ for 7–10 days. Two clean 6 mm sapphire disks (Technotrade Inc.) were placed per well of 12-well tissue culture plate and coated with poly-D-lysine (1 mg/ml, Sigma) and collagen (ThermoFisher). Astrocytes serve as a feeder layer for neurons and were seeded (50,000/well) 1 week before hippocampal neuronal culture. On day 6, FUDR (80 µM) was added to inhibit cell division. On day 7, hippocampi were isolated and digested with papain (20 U/ml) for 30–60 min at 37 °C with shaking. An inactivation solution (2.5 mg of trypsin inhibitor and 0.5 mg of albumin per mL of DMEM) was then applied for 5 min at 37 °C. Hippocampi were triturated by pipetting 4 × 20 times, and cells were seeded on prepared astrocyte feeder layer with a density of 75,000/well and maintained in Neurobasal A (NBA) media supplemented with B27, Glutamax, and 0.2% penicillin-streptomycin at 37 °C and 5% $CO_2$. The cells were infected with lentivirus at days in vitro (DIV) 3 and 4 as needed and used for experiments on DIV 13–17.

**Expression constructs**. Lentiviral expression constructs were used to express transgenes in neurons. All vectors were based on the lentiviral shuttle vector FUGW[68]. Gria2 cDNA was N-terminally tagged with HaloTag and cloned in frame downstream of synapsin-1 promoter with NLS-GFP-P2A (NGP) sequence (a gift from Christian Rosenmund lab). The NLS-GFP signals were used to evaluate viral infection and intensity of transgene expression from this polycistronic construct. Nucleotides corresponding to hexa-histidine residues were inserted in Gria2 sequence downstream of the signal sequence using Gibson cloning strategy and NEB builder kit (New England Biolab) to generate NGP-(His)6-GluA2. To generate NGP-SnapTag::GluA2, the hexa-histidine tag was replaced with a Snap-Tag by In-Fusion cloning (Takara Bio). pCI-SEP-NRI was a gift from Robert Malinow (Addgene plasmid # 23999; http://n2t.net/addgene:23999; RRID: Addgene_23999)[69]. Nucleotides corresponding to hexa-histidine were inserted downstream of the signal sequence of NR1 using Gibson cloning strategy and NEB builder kit (New England Biolab) followed by sub-cloning into the lentiviral shuttle vector FUGW to obtain NGP-(His)6-NR1. To generate NGP-SnapTag-NR1, the hexa-histidine tag was replaced with a SnapTag by In-Fusion cloning (Takara Bio).

**Lentivirus production and infection**. Lentiviruses carrying the expression constructs were produced using the following procedures. The bottom surface of T-75 flasks was coated with poly-L-lysine (0.2% in milliQ water). A day before the transfection, HEK293T cells were plated at $6.5 × 10^5$/ml (10 ml in T-75) in NBA media containing 1% glutamax, 2% B27, and 0.2% penicillin-streptomycin. The shuttle vector (FUGW)[68] containing expression constructs and helper plasmids (VSV-G and CMV-dR8.9) was mixed at 20, 5, and 7.5 µg, respectively, in 640 µl NaCl solution (150 mM) (solution I). Another solution (solution II) was prepared as follows: 246.7 µl $H_2O$, 320 µl NaCl (300 mM), 73.3 µl polyethylenimine (0.6 µg/µl). Solutions I and II were mixed by vortexing and incubated at room temperature for 10 min, followed by addition to the T-75 flask containing HEK293T cells. The cells were incubated at 37 °C (5% $CO_2$), and the viruses were harvested 3 days later. The media containing lentivirus was centrifuged at 2880 × g to obtain 20-fold concentration using Amicon (Ultracel-100k). The infection efficiency was determined by infection in wild-type neurons that were separately prepared. For all the experiments, dissociated hippocampal neurons were infected on DIV 3 and 4 with lentiviruses carrying the expression constructs. The infection rate of 95% was achieved in all cases.

**Ni-NTA-gold labeling**. For gold labeling, solution containing Ni-NTA-gold (Nanoprobes) was added to each well containing two sapphire disks so that the final concentration of 5 nM is achieved. Cells were incubated for 30 min in the $CO_2$ incubator set at 37 °C. Each disk was washed thoroughly by agitating in a small petri dish (30 mm) containing physiological saline solution (140 mM NaCl, 2.4 mM KCl, $CaCl_2$ 1.2 mM, and 3.8 mM $MgCl_2$, 10 mM HEPES, 10 mM glucose; pH adjusted to 7.3 with NaOH, 300 ± 5 mOsm). The washing procedure was repeated three times to minimize background labeling. Immediately after washing, cells were mounted for high-pressure freezing.

**High-pressure freezing and freeze-substitution**. Cells cultured on sapphire disks were frozen using a high-pressure freezer (EM ICE, Leica Microsystems). Following gold labeling, each disk with neurons was transferred into the physiological saline solution containing NBQX (3 µM, Tocris) and bicuculline (30 µM; Tocris), which were added to block recurrent synaptic activity during the zap-and-freeze experiments. The disk was mounted onto the photoelectric middle plate with neurons facing up. A 100 µm spacer ring was placed on top of the sapphire disk. Then, another blank sapphire disk was placed on top of the spacer ring to make a "sandwich." Finally, a rubber ring was put on top of the "sandwich" to hold it in place. The entire assembled middle plate was then placed on a piece of filter paper to remove the excess liquid, loaded between two half cylinders, and transferred into the freezing chamber. An electrical field of 10 V/cm was applied for 1 ms to induce a single action potential, and cells were frozen 5 and 11 ms after the stimulus[4]. These time points are chosen based on our recent study suggesting that pits captured at these time points represent fusion intermediates during synchronous and asynchronous release, respectively[4]. The exact proportion of these events could not be determined, but based on EGTA experiments, asynchronous release may account for up to 20% of the currents in these synapses[70,71]. A 1-ms electrical pulse likely induces an action potential from all neurons on a disk uniformly, and ~35% of synapses exhibit fusion pits[4]—consistent with the synaptic release probability of these synapses[72–76]. For no stimulation control, the photoelectric middle plate was programmed not to discharge. The frozen sample was automatically dropped into a storage dewar filled with liquid nitrogen.

After freezing, the middle plate with sapphire disks was transferred to a cup containing anhydrous acetone (−90 °C), which was placed in an automated freeze-substitution system (EM AFS2, Leica microsystems) using prechilled tweezers. The cryovials containing fixative (1% glutaraldehyde, 1% osmium tetroxide, and 1% water in anhydrous acetone) were stored in liquid nitrogen and moved to AFS2 before use. After disassembling the freezing apparatus, sapphire disks with neurons were transferred into cryovials in the AFS, which is set at −90 °C, using prechilled tweezers. The freeze-substitution program was as follows: −90 °C for 6–10 h, −90 to −20 °C in 14 h, −20 °C for 12 h, and −20 to 20 °C in 4 h.

**Embedding, sectioning, and transmission electron microscopy.** Following freeze-substitution, fixatives were washed with anhydrous acetone for three times, 10 min each. After washing, samples were infiltrated through 30, 70, and 90% epon araldite in anhydrous acetone every 2 h. Then samples were transferred to caps of polyethylene BEEM capsules with 90% epon araldite and incubate overnight at 4 °C. Next day, samples were incubated in the caps of polyethylene BEEM capsules with 100% epon araldite (epon 6.2 g, araldite 4.4 g, DDSA 12.2 g, and BDMA 0.8 ml) at room temperature. Samples were transferred to new caps with fresh 100% epon araldite every 2 h three times, after which samples were baked at 60 °C for 48 h.

After resin was cured, sapphire disks were removed from resin. Cells were embedded in the resin block. Then, the block was cut into small pieces and place atop of a dummy block using super glue for sectioning. 40 nm sections were cut using an ultramicrotome (EM UC7, Leica microsystems) and collected on single-slot copper grids coated with 0.7% pioloform. The sections were stained with 2.5% uranyl acetate in 75% methanol and then imaged at 80 kV at the ×93,000 magnification on a Philips CM120 transmission electron microscope equipped with an AMT XR80 camera. Images are acquired through AMT Capture v6. About 100 electron micrographs per sample were taken blindly. To avoid the sampling bias, synapses were found by bidirectional raster scanning along the section at ×93,000. Synapses were identified by a vesicle-filled presynaptic bouton and a PSD.

**Spin-mill serial block face imaging.** Samples were sent to ThermoFisher for spin-mill serial block face imaging. Spin-milling experiments were performed on a DualBeam instrument (ThermoFisher Scientific, Helios Hydra). A whole resin block (6 mm diameter) was glued onto a scanning electron microscope (SEM) stub using silver conductive epoxy without pre-trimming or sputter coating. The sample was positioned at the eucentric position of the system, and the stage was tilted to −34 degrees, such that the focused ion-beam (FIB) angle of incidence was 4 degrees from glancing relative to the sample surface.

The spin-milling process consists of the following sequence. First, oxygen FIB beam (12 keV, 65 nA) was applied in a $400 \times 100$ μm box pattern on desired sample area for 10 s (dwell time set at 200 ns) to expose a new surface of the sample. The stage was compucentrically rotated 60 degrees, and the sample milled again with another FIB exposure. This process was repeated six times to achieve a full 360-degree rotation of the sample. Ion flux was delivered to the sample from several different azimuthal directions to reduce textural artifacts generated during the ion milling. One full rotation of milling constituted a "z slice." Second, the sample was tilted back to a stage tilt of zero degree to perform SEM imaging. Images were collected from multiple regions-of-interest. These two steps were automated with Autoscript software (ThermoFisher Scientific) and repeated until the desired volume of images was collected—similar to the serial block face imaging technique[77]. The milling slice thickness was controlled to achieve 20 nm. Ten areas of interests were acquired in parallel with a X, Y pixel size of 1 nm. A total of ~40 slices was collected from each sample. The resulting 3D data sets were aligned and visualized using the Amira software (ThermoFisher Scientific).

**Electron microscopy image analysis.** All the images from a single experiment were shuffled for analysis as a single pool using a custom R (R Development Team, R Studio 1.3, R version 3.5.1) script. Images that could not be reliably segmented, either because the image was not of a bona fide synapse or morphology was too poor, were excluded from segmentation; this was done only after randomizing the images. No other data were excluded. The plasma membrane, active zone, PSD, docked synaptic vesicles, synaptic vesicles close to the active zone, pits (putative fusion events), and gold particles were annotated in Fiji (version 1.0) using a custom macro. The active zone was identified as the region of the presynaptic plasma membrane juxtaposed to the PSD. Docked vesicles were identified by their membrane appearing to be in contact with the plasma membrane at the active zone (0 nm from the plasma membrane), that is, there are no lighter pixels between the membranes. Pits were identified as smooth curvature (not mirrored by the post-synaptic membrane) in an otherwise straight membrane. These pits are considered exocytic[4]. Pits outside the active zone are considered endocytic or membrane ruffles, as this is the primary site for ultrafast endocytosis[50]. Under these criteria, we could miss or over-annotate vesicles and pits. To minimize the bias and maintain consistency, all image segmentation, still in the form of randomized files, was thoroughly checked by a second member of the lab. However, no corrections were made for synaptic vesicles since vesicles are much more abundant, and the same criteria were used to annotate them in all conditions. A similar amount of overestimate is expected in this case. Features were then quantitated using custom MATLAB (MathWorks R2017-R2020a) scripts (available from: https://github.com/shigekiwatanabe/SynapsEM).

Location of pits, docked vesicles, and gold particles within the active zone/PSD from single sections was calculated from the distance from the center of the pit to the center and the edge of the active zone in 2D. Distance from the center was normalized by dividing the distance to the edge by the half-width of the active zone. For 3D data, the distance to the center of the active zone was calculated from serial sections. First, the location in 2D was calculated as above. Then, the 3D distance was calculated to the center of the active zone in the middle section of the series using the Pythagorean theorem with the assumption that each section is the same thickness and the center of the active zone aligns in each image. Locations in

3D data were further corrected to be the density of vesicles/pits at each distance from the center of the active zone or PSD. To calculate density of vesicles, pits, and gold particles from the center to the edge in 3D reconstructions, the radial position of each vesicle/pit/gold particle was converted to the fractional area of a circle bounded by that radius. In the case of a unit circle (distance from center to edge is by definition 1 when data normalized to the size of the PSD), this is simply the square of the original normalized distance to the center. Example micrographs shown were adjusted in brightness and contrast to different degrees (depending on the varying brightness and contrast of the raw images), rotated, and cropped in Adobe Photoshop.

**Cluster analysis.** Sequential annotated electron microscopy slices of gold-labeled NMDA and AMPA receptors were used to generate a 3D spatial map of receptors for each synapse (MATLAB 2019b, Mathworks). For each synapse, K-means clustering (Lloyd's algorithm) was performed for $1$–$N$ number clusters, where $N$ is the total number of gold-labeled receptors. The optimal number of clusters was obtained by calculating the knee-point of the within-cluster sum of square differences (SSD) as a function of number of clusters. A final generated synapse with the optimized number of clusters was then rendered over each synapse for both AMPA and NMDA receptors. These spatial maps allow visualization and measurement of the locations of each receptor with respect to the center of the synapse. To determine whether the clustering of these receptors was due to chance, for each mapped synapse and respective number of particles, we generated 50 maps with randomized particle positions using custom scripts. The above K-means clustering paradigm was then run on all 50 maps and the mean SSD was recorded for each synapse. 2D scatter plots of SSD and synaptic sizes for the experimental and randomized condition for AMPA and NMDA receptors were generated, where low SSD indicates more tightly packed clusters. Linear regression analysis (Prism 7 and 8.2.0, GraphPad) was run for each set of conditions to determine whether the experimentally clustered receptors were significantly different than randomized clusters.

**Computer simulations.** Computer modeling was performed using the MCell/CellBlender simulation environment (mcell.org) version 3.5.0-9 for Linux. The realistic model of glutamatergic synaptic environment was constructed from 3D electron microscopy of hippocampal area CA1 neuropil[27,78,79]. The kinetic scheme and kinetic rate constants for AMPAR (GluAR) activation and desensitization by glutamate were obtained from previously published reports (see ref. [51] for details of the kinetic scheme and ref. [37] for the rate constants). The NMDAR kinetics were obtained from Vargas-Caballero and Robinson[39]. Since the time course of the diffusion and presence of glutamate in the synaptic cleft is especially important in the model presented here, our model also included realistic extracellular space a kinetic model of glutamate transporters distributed on astrocytic glial processes in the surrounding neuropil[27].

The initial distribution of AMPA and NMDA receptors as well as the location of the presynaptic neurotransmitter release site was established by running a dynamic simulation to allow self-organization of the distributions. To accomplish this, two surface properties were defined: the synapse and the PSD (identified on electron microscopy data). According to the literature 200 PSD-95 molecules, 60 AMPA receptors, and 30 NMDA receptors were available on the spine head. These molecules were allowed to freely diffuse at the synapse. Inside the PSD, PSD-95 was reversibly palmitoylated (pPSD-95) at a defined rate (kon = 35, koff = 0.7).

A clusterization point called "L" was placed at the center of the PSD. pPSD-95 aggregates in contact with L (kon = 7, koff = 1) to form a domain. Mobile NMDA receptors interacted with this domain and were trapped into an NMDA receptor cluster (kon = 10, koff = 1). A mobile "G" molecule was released inside the PSD and was immobilized at random location when it randomly interacted with a PSD-95. After immobilization, the molecule of "G" recruited the insertion of a presynaptic neurotransmitter release site into the presynaptic membrane at the point closest to the location of G. At the same time, "G" clustered pPSD-95 (kon = 100, koff = 1), which, in turn, clustered AMPA receptor (kon = 10, koff = 1).

The approach to a final steady-state organization of AMPA receptor and NMDA receptor at the synapse was simulated with a time step of 1 ms for 10,000 iterations (10 s), until reaching a steady. It is important to note that the means employed here to achieve receptor organization is only intended to give the desired final organization in our model and is not intended to model the physiological mechanisms by which this occurs in real synapses. After reaching the desired organization, the simulations were switched to a time step of 1 μs for 250,000 iterations to model the AMPA receptor and NMDA receptor and when the glutamate was released at the presynaptic level, in front of the "G" aligned with AMPA receptor or in front of "L" aligned with the NMDA receptor.

After binding neurotransmitter, the flux of ionic current through activated NMDAR is voltage dependent due to channel blockade by $Mg^{2+}$ at hyperpolarized membrane potential[40]. We simulated this voltage-dependent blockade with the following approach[27]. The neural simulation program NEURON was used to simulate excitatory postsynaptic potentials of the desired timing and amplitude in the spine head located on the dendritic branch of a modeled pyramidal neuron. The time-varying voltages recorded at the spine during these stimuli were used to drive voltage-dependent transition rates in the model of NMDA receptor activation kinetics. The membrane potential waveform simulated following a single vesicle

release is consistent with the previous estimate of voltage change in spines[80,81]. The voltage changes alter the kinetic constants for relief of the $Mg^{2+}$ block of the NMDA receptors and determine the ion current flux through open NMDA receptors. Each individual NMDA receptor channel opens and fluxes current in the simulation only when glutamate is bound to the receptor at the time that the $Mg^{2+}$ block is relieved. MCell uses stochastic Monte Carlo methods and simulation results reflect the realistic behavior of stochastic channel fluctuations. For all computational experiments, 48 trials were performed, allowing estimation of the mean and standard deviation of the time course of channel activation. Data were analyzed using custom Python scripts using Python-3.4, NumPy-1.9.2, SciPy-0.15.1, and matplotlib-1.4.3.

**Statistical analysis**. All data showing distribution of receptors, vesicles, and pits are pooled from multiple experiments. The number data shown are per experiment. All data were initially examined on a per experiment basis (with all freezing done on the same day and all segmentation done in a single randomized batch); none of the pooled data show any result that was not found in each replicate individually. We did not predetermine sample sizes using power analysis, but based them ($N = 2$ and 3, $n > 200$) on our prior experience with flash-and-freeze data[50,82,83]. An alpha of 0.05 was used for statistical hypothesis testing. All data were tested for normality by D'Agostino–Pearson omnibus test to determine whether parametric or nonparametric methods should be used. Comparisons between two groups were performed using a two-tailed Welch two-sample $t$-test or Wilcoxon rank-sum test. Comparisons between multiple groups followed by full pairwise comparisons were performed using one-way analysis of variance followed by Tukey's HSD test or Kruskal–Wallis test followed by Dunn's multiple comparisons test. For testing whether locations of pits or receptors were biased toward the center or edge of the synapse, a two-tailed one-sample $t$-test or Wilcoxon rank-sum test with a theoretical median of 0.5 was used (each of these $p$ values, as well as that of the comparisons between pit locations in different samples, were accordingly corrected for multiplicity using Bonferroni's method). All statistical analyses were performed and all graphs created in Graphpad Prism.

**Life Sciences Reporting Summary**. More details on experimental procedures, materials, and statistics are available in the Life Sciences Reporting Summary.

## Data availability

All data supporting the findings of this study are provided within the paper as source_data.xlsx and its Supplementary information. Full data tables underlying the figures are available at: https://figshare.com/projects/Asynchronous_release_sites_are_aligned_with_NMDA_receptors_in_mouse_hippocampal_synapses/87923. Raw images and image analysis files are available upon request. Source data are provided with this paper.

## Code availability

Custom R and MATLAB scripts are available through https://github.com/shigekiwatanabe/SynapsEM[84,85].

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

## Acknowledgements

We are indebted to Quan Gan, Kie Imoto, and Chengxiu Zhang for cell culture, help with gold labeling experiments, and stimulating discussions. We also thank Mike Delanoy and Barbara Smith for technical assistance with electron microscopy. We thank Cveta Tomova, Chad Rue, and Kenny Mani at ThermoFisher Scientific for facilitating the collaboration on Helios Hydra DualBeam, and Christian Rosenmund and M.M.B. for sharing unpublished data. We also thank the Marine Biological Laboratory and their Neurobiology course for supporting the initial set of experiments (course supported by National Institutes of Health grant R25NS063307). S.W. and this work were supported by start-up funds from the Johns Hopkins University School of Medicine, Johns Hopkins Discovery funds, and the National Science Foundation (1727260), the National Institutes of Health (1DP2 NS111133-01 and 1R01 NS105810-01A1) awarded to S.W. S.W. is an Alfred P. Sloan fellow, McKnight Foundation Scholar, and Klingenstein and Simons Foundation scholar. G.K. was supported by a grant from the National Institutes of Health to the Biochemistry, Cellular and Molecular Biology Program of the Johns Hopkins University School of Medicine (T32 GM007445) and is a National Science Foundation Graduate Research Fellow (2016217537). E.H. and T.M.B. are supported by CRCNS-NIH-ANR grant AMPAR-T. The EM ICE high-pressure freezer was purchased partly with funds from an equipment grant from the National Institutes of Health (S10RR026445) awarded to Scot C Kuo.

## Author contributions

S.W. and S.L. designed the experiments and analyzed the data, and wrote the paper. S.L., S.R., and G.K. performed all freezing experiments and single-section electron microscopy sample preparation, imaging, and analysis, with technical assistance from S.W. J.W. performed the spin-mill imaging, and S.L., S.A.L., C.P., and S.S. analyzed the 3D data. S.W. and S.A.L. developed MATLAB code for image analysis. M.M.B. designed the SMASH approach. H.F. and R.R. performed pilot experiments and electron microscopy sample preparation, imaging, and analysis. S.W., T.M.B., and E.H. developed and performed computer simulations. S.W. funded and oversaw the research.

## Competing interests

The authors declare no competing interests.
