## [Peer Review File · Nature Communications]

Reviewers' Comments:

Reviewer #1:

Remarks to the Author:

The manuscript by Li et al. reports the position of AMPARs and NMDARs within the postsynaptic membrane and in relation to the position of docked as well as fusing synaptic vesicles (SVs). For this the authors have used an elegant combination of gold-labelling of his-tagged glutamate receptors and high-pressure freezing and freeze substitution of time-resolved electrically stimulated cultured neurons. They show that AMPARs are more concentrated at the edge of the active zone than at its center. NMDARs are more evenly distributed. Synchronously fusing SVs (observed 5ms after depolarization) are fairly evenly distributed throughout the active zone. On the other hand asynchronously fusing SVs (observed 11 ms after depolarization) are concentrated at the center of the active zone. Consequently AMPARs are significantly closer to synchronously fusing SVs than to asynchronously fusing SVs, whereas the opposite relation is observed for NMDARs. Then the authors simulated NMDAR response as a function of the time delay between synchronous and asynchronous, as well as of the order of receptor type activation (e.g. first AMPAR then NMDAR or vice versa). They conclude that the alignment of the first release site with AMPAR ensures maximal depolarization and ensures efficient activation of NMDAR. Finally they present a model suggesting what influences the localization of synchronous vs asynchronous release sites.

The techniques are original and the results are very interesting for neuroscientists at large. Statistical analysis is sound. Yet there are some issues that should be addressed:

- The manuscript builds upon and refers to work that has not yet been published in a peer-reviewed paper but is only deposited on biorxiv (« Our recent study suggests that release sites for synchronous and asynchronous release occupy unique domains within an active zone: synchronous release sites are uniformly distributed, while asynchronous release sites are abundant near the center of an active zone (REF 4). However, the functional importance of this spatial organization is unknown. »). As the correctness of the results in the preprint are essential for the validity of the present manuscript, I would like to have REF 4 accepted in a peer reviewed journal before making a final judgement on the present manuscript.
- The authors make an analysis of spin-mill serial block face images, which is quite important for the conclusions of the manuscript. Yet no raw images or stacks are shown. Showing some images and movies would be helpful.
- The authors report NMDAR distribution both from 2D and 3D image analysis. They claim that NMDARs are uniformly distributed at the surface of postsynaptic density. In the 2D plot (fig. 1d), I get to the same conclusion. Yet the 3D reconstruction shown in fig. 1e and the plot in fig. 1f does not show a uniform distribution of NMDARs but a skew towards the center of the PSD. According to the latter plot half of NMDARs are located closer than 0.35 to the PSD center (relative distance on a scale from 0 to 1). How can the authors explain this discrepancy?
- In figure 3a,c,and e the number of activated NMDARs is plotted. How many NMDARs were included in the simulation?
- The cartoon in fig. 3g does not correspond to some of the interpretations of the results in the text. First, NMDAR are almost exclusively concentrated at the center of the PSD. In the text they state that NMDARs are evenly distributed. Second, in the text they state : « exocytotic pits at 5ms were distributed throughout the active zone with a slight bias to the center, while fusion intermediates at 11 ms were strongly biased towards the center. » On the cartoon they draw calcium channels at the edge of the active zone and initial fusion pits next to these channels. This does not fit with the statement above.

- This last point brings another issue: how can we explain that the very first fusion events are distributed throughout the active zone whereas late when they are supposed to take place just next to calcium channels?

Benoit Zuber

Reviewer #2:

Remarks to the Author:

The present MS describes experiments that localize AMPA and NMDARs at high resolution in cultured hippocampal synapses. The authors conclude that these distinct types of GluRs have distinct subsynaptic distributions, which has been known for >20 years. The authors also investigated the spatial relationship between these receptors and exocytic pits and concluded that AMPARs are closer than NMDARs to synchronously released vesicles.

The introduction of the MS is rather biased towards a view that has supporting evidence, but totally ignores a much larger body of work that is inconsistent with this view. One set of published data support the view that the magnitude of the postsynaptic AMPAR-mediated EPSC depends on the nano-arrangement of the location of the vesicle fusion and the clusters of postsynaptic AMPA receptors, whereas there are a large body of experimental and modelling work arguing for a large postsynaptic receptor occupancy and the insignificance of the exact location of the vesicle fusion. The reviewer insists that the Introduction must contain both views, as this issue is far from being concluded.

The authors also seem to forget a large body of data from the 90's that examined the sub-PSD distribution of AMPA and NMDARs in many brain areas, including the striatum, cortex, hippocampus etc.

The argument of the need of temporal precision for NMDA receptor activation is also biased! The time to first opening distribution of NMDARs is tens of milliseconds. The diffusion/equilibration of glutamate within the cleft is tens of microseconds. The reviewer cannot see why 100 us delay in the activation of NMDARs 200 nm away would have any significant effect if the receptors open 20 ms later?

The exact proportion of synchronously and asynchronously released vesicles is unknown in this synapse. However, it is well known that most vesicles are release synchronously, unlike in some other synapses (e.g. CCK expressing GABA cells). Thus, the functional consequences of the role of asynchronously released vesicles and their role in NMDA receptor activation remain unknown until this proportion is precisely determined. Why do the authors believe that the arbitrary 5 ms reveals synchronously, whereas 11 ms asynchronously released vesicles? Is the probability of finding such pits consistent with the probability of vesicle release? Some of these synapses release vesicles with a probability of <1%! How do the authors handle such synapses?

The authors took a novel approach to label GluA2 containing receptors (His-tag and affinity labelling). They report that 70% of the overexpressed synapses has ~2-3 gold particles. The exact density of AMPA receptors in excitatory synapses is unknown. However, a lower limit can be estimated from the highest labeling achieved so far, that is ~1200 gold/um². If hippocampal synapses are ~0.05-0.1 um², a minimum of 60-120 AMPARs should be present, out of which the authors label 2-3. This low labeling efficacy also affects the outcome of the clustering. Have the authors considered what effect would it have if they analyzed synapses with 100 rather than 10 gold?

In Figure 1, the authors present one image with which they illustrate that gold particles for AMPARs are located towards the periphery of the synapse. In suppl figure 1 (where they 'validate' the specificity of their method) they provide 4 electron micrographs of 4 synapses in which gold particles for AMPARs are rather uniformly distributed; there are just as many golds in the middle than at the edge.

The identification of the fusion events in the EMs is rather uncertain (Fig 2a, b). When one looks at the presented micrographs, the membranes are totally disintegrated with large, swollen extracellular spaces surrounding the synapses (which are impossible to identify based on the

presented images!).

In such EM images, the identification of docked vesicles and exocytic pits is problematic, to say the least! The authors do not provide data about the density of docked vesicles in their AZ, which would allow direct comparisons to published data using chemical fixation or only high pressure freezing. Without knowing that, a median distance of 100 nm from the docked vesicles to the AMPA/NMDARs indicate a maximum of 2-3 docked vesicle per AZ. That is much lower than estimates using the best methods (e.g. Imig et al).

The authors interpretation of their own data is interesting: 'Of note, exocytic pits at 5 ms were distributed throughout the active zone with a slight bias to the center (median = 0.4)' few sentences below: 'These results suggest that neurotransmitter is released synchronously near AMPA receptors and asynchronously around NMDA receptors.'

Reviewer #3:

Remarks to the Author:

This is an interesting paper combining cutting edge EM and very nice Monte Carlo simulations to examine the implications of distinct subsynaptic localization of AMPA and NMDA receptors (GluARs and GluNRs). The authors show that synchronous neurotransmitter release occurs in regions of the active zone (AZ) that are directly apposed by GluARs in the postsynaptic density (PSD), whereas GluNRs are localized in distinct clusters that face sites that release asynchronously. The modeling then addresses whether this arrangement permits "priming" of GluNRs (i.e., occupancy of one of the two glutamate binding sites during one release event to facilitate greater activation during the next release event), and also whether it maximizes the capacity of GluARs to depolarize the postsynaptic membrane and relieve the Mg block on GluNRs.

The impact of this study rests upon its ability to overturn about 30 years of thinking in the field that GluARs and GluNRs are effectively co-localized in the synaptic cleft, sufficiently activated by a single vesicle of glutamate, and activated in similar proportions during spontaneous and evoked release (Bekkers and Stevens, 1989, Nature; Bekkers, et al., 1990, PNAS). It also calls into question the even longer-lived tenet, back to Bernard Katz, that evoked responses comprise a slightly asynchronous collection of events accurately represented by those recorded spontaneously as mEPSCs. The MCell modeling addresses rather exotic ideas at the expense of simpler, more fundamental predictions that can be tested experimentally:

First, in Mg-free solutions do mEPSCs exhibit a larger GluNR component (relative to the GluAR component) than evoked EPSCs?

Second, do GluNRs see less glutamate than GluARs during an evoked EPSC, and is the opposite the case during mEPSCs? This is a harder question to address experimentally (but see Liu, et al., 1999, Neuron).

Given what has come before, the strong expectation is that GluARs and GluNRs are activated to similar relative extents during miniature and evoked EPSCs. If that is, in fact, true, then the anatomical results shown here would appear to reflect a molecular ultrastructure that organizes postsynaptic receptors via distinct mechanisms that nonetheless enables them to be activated similarly in response to spontaneous or evoked release. It is reasonable for the authors to suggest that such electrophysiological studies are beyond the scope of the current study, but their omission diminishes the impact substantially.

p. 20: in the methods, the authors note that the GluAR kinetic parameters used in the simulations were adjusted to fit with recorded mEPSCs. This seems circular. One would presume that this is accomplished by placing the GluARs at some distance from an asynchronous release site and then adjusting the kinetics to match mEPSCs recorded by others. The GluAR kinetic properties were obtained in excised patches (from CA3 pyramidal cells) and so provide good measures of binding

and unbinding rates regardless of a particular glutamate waveform. If major adjustments were required, it seems more likely to suggest that the proposed glutamate waveform reaching GluARs during an asynchronous event may be incorrect.

Minor comment:

p. 11: the point about GluAR binding affinity being low is potentially misleading here, as affinity is measured at equilibrium, and a synaptic event is far too brief to approach equilibrium. In the 100 us of a synaptic event, the activation of postsynaptic receptors depends primarily on their glutamate binding rate, which is actually higher for GluARs than for GluNRs.

Point-by-point response to reviewers

Nature communication manuscript: NCOMMS-20-19533-T

Reviewers' comments *italicized*.

Reviewers' comments have been renumbered for easy reference.

Responses are labeled.

Reviewer #1 (Remarks to the Author):

1. The manuscript by Li et al. reports the position of AMPARs and NMDARs within the postsynaptic membrane and in relation to the position of docked as well as fusing synaptic vesicles (SVs). For this the authors have used an elegant combination of gold-labelling of his-tagged glutamate receptors and high-pressure freezing and freeze substitution of time-resolved electrically stimulated cultured neurons. They show that AMPARs are more concentrated at the edge of the active zone than at its center. NMDARs are more evenly distributed. Synchronously fusing SVs (observed 5ms after depolarization) are fairly evenly distributed throughout the active zone. On the other hand asynchronously fusing SVs (observed 11 ms after depolarization) are concentrated at the center of the active zone. Consequently AMPARs are significantly closer to synchronously fusing SVs than to asynchronously fusing SVs, whereas the opposite relation is observed for NMDARs. Then the authors simulated NMDAR response as a function of the time delay between synchronous and asynchronous, as well as of the order of receptor type activation (e.g. first AMPAR then NMDAR or vice versa). They conclude that the alignment of the first release site with AMPAR ensures maximal depolarization and ensures efficient activation of NMDAR. Finally they present a model suggesting what influences the localization of synchronous vs asynchronous release sites.

The techniques are original and the results are very interesting for neuroscientists at large. Statistical analysis is sound. Yet there are some issues that should be addressed:

Response: We thank Dr. Zuber for the careful read of our manuscript and positive outlook. The point-by-point responses are found below.

2. The manuscript builds upon and refers to work that has not yet been published in a peer-reviewed paper but is only deposited on biorxiv (« Our recent study suggests that release sites for synchronous and asynchronous release occupy unique domains within an active zone: synchronous release sites are uniformly distributed, while asynchronous release sites are abundant near the center of an active zone (REF 4). However, the functional importance of this spatial organization is unknown. »). As the correctness of the results in the preprint are essential for the validity of the present manuscript, I would like to have REF 4 accepted in a peer reviewed journal before making a final judgement on the present manuscript.

Response: The reference 4 is now accepted at Nature Neuroscience.

3. The authors make an analysis of spin-mill serial block face images, which is quite important for the conclusions of the manuscript. Yet no raw images or stacks are shown. Showing some images and movies would be helpful.

Response: As suggested, the raw images are included as movies and available as Supplementary Movies 1-4.

4. The authors report NMDAR distribution both from 2D and 3D image analysis. They claim that NMDARs are uniformly distributed at the surface of postsynaptic density. In the 2D plot (fig. 1d), I get to the same conclusion. Yet the 3D reconstruction shown in fig. 1e and the plot in fig. 1f does not show a uniform distribution of NMDARs but a skew towards the center of the PSD. According to the latter plot half of NMDARs are located closer than 0.35 to the PSD center (relative distance on a scale from 0 to 1). How can the authors explain this discrepancy?

Response: Dr. Zuber is correct. In the 3D reconstruction, NMDA receptors are enriched near the center. This point was noted in the original manuscript.

p. 7: “Similar to single profiles, AMPA and NMDA receptors were differentially distributed in the reconstructed postsynaptic densities (**Fig. 1f**, $p < 0.001$). As in single profiles (**Fig. 1d**), AMPA receptors were biased towards the edge (**Fig. 1f**, $p < 0.001$, **Supplementary Fig. 2b** for more example). Interestingly, NMDA receptors were enriched around the center of the postsynaptic density (**Fig. 1e,f**, $p < 0.001$, **Supplementary Fig. 2d** for more example). The distributions of these receptors are consistent with previous experiments using single-molecule localization microscopy^{33,35} and electron microscopy^{14,49} .

However, as pointed out, the distributions in single profiles and 3D reconstructions were different, although they both show the same tendency (AMPA receptors are towards the edge, and NMDA receptors towards the center). What accounts for the discrepancy?

A simple answer is that single synaptic profiles are random representations of synapses; each image represents a 40-nm segment of a synapse, sliced at a random location. These slices are not necessarily going through the center of an active zone/post-synaptic density. Thus, although NMDA receptors are localized near the center of a post-synaptic density when reconstructed, they may not appear enriched towards the center in individual images. Likewise, AMPA receptors are distributed towards the edges of a reconstructed post-synaptic density, but gold particles may be found near the center of a post-synaptic density in given images (**Supplementary Fig. 1a**). Thus, the distributions determined from single profiles would appear different from those from the 3D reconstruction.

Nonetheless, the data from single profiles can be used to approximate the distribution when sufficient data are collected. To draw this point clear, we used the datasets from the 3D reconstruction and measured the distribution of receptors to the center of a postsynaptic density in each image, instead of the true center in the reconstructed postsynaptic density (**Supplementary Fig. 2a**). When we map the first 25 gold particles, the distributions of both AMPA and NMDA receptors are nearly identical (**Supplementary Fig. 2a**), with their means being both 0.55. This is likely because we are sampling sparsely. With 100 gold particles mapped, the means are 0.54 and 0.44, respectively, which are similar to the means obtained from all particles mapped (0.57 and 0.42). Qualitatively, these data are quite similar to the data obtained from the 3D reconstructions (means: 0.56 and 0.33) and single profile data obtained from random synapses (means: 0.6 and 0.5). Thus, although true distributions of gold particles cannot be determined from single profiles, they can be estimated from a large dataset.

To make these points clear, we added the following to the manuscript.

p. 6, “note that synaptic profiles are random 40-nm segments of synapses”

p.7, “However, single synaptic profiles are random representations of synapses; each image represents a 40-nm segment of a synapse, sliced at a random location. These slices are not necessarily going through the center of an active zone/post-synaptic density. Nonetheless, the data from synaptic profiles can be used to estimate the distributions when sufficient data are collected (Supplementary Fig. 2a).”

5. In figure 3a,c,and e the number of activated NMDARs is plotted. How many NMDARs were included in the simulation?

Response: We placed 15 NMDA receptors around the center of the postsynaptic density based on the literature (Spruston, Jonas, and Sakmann, 1995; Takumi et al., 1999), which is similar to the estimate in the current manuscript (~10 NMDA receptors/synapse).

p. 10-11 “We further incorporated the observed distributions of receptors and their numbers revealed by superresolution imaging³⁵ (**Fig. 1**): clusters containing ~20 AMPA receptors and ~15 NMDA receptors were placed around the periphery and center of the postsynaptic density, respectively.”

6. The cartoon in fig. 3g does not correspond to some of the interpretations of the results in the text. First, NMDAR are almost exclusively concentrated at the center of the PSD. In the text they state that NMDARs are evenly distributed. Second, in the text they state : « exocytotic pits at 5ms were distributed throughout the active zone with a slight bias to the center, while fusion intermediates at 11 ms were strongly biased towards the center. » On the cartoon they draw calcium channels at the edge of the active zone and initial fusion pits next to these channels. This does not fit with the statement above.

Response: We thank Dr. Zuber for bringing up these points. We agree that some drawings are hypothetical and may be misleading. In particular, calcium channels were arbitrary placed near the fusing vesicles and excluded from the center of an active zone in the cartoon. This was based on the data that 1) vesicles fuse synchronously near the calcium channels and 2) asynchronous release, which is concentrated near the center, is unlikely coupled to calcium channels. However, we did not demonstrate the locations of calcium channels in this study. Thus, we removed the calcium channels from the model.

For the initial fusion event, we placed the pit towards the periphery in the original model to better represent the tight alignment between synchronous release sites and AMPA receptors, since AMPA receptors are enriched near the edge. However, it is correct that the distribution of fusion events was slightly biased towards the center in the single profile analysis. Thus, we shifted the location of the initial fusion event slightly towards the center from the original location.

However, regarding the NMDA receptor distribution, we think the cartoon represents the data quite accurately for two reasons. First, in the serial-section reconstruction data, NMDA receptors are strongly enriched near the center of a postsynaptic density (Fig. 1f), although NMDA receptors appear uniformly distributed in single profiles (Fig. 1d). Thus, in the three-dimensional model, they are likely distributed towards the center. Second, a recent paper from Dr. Hossy also demonstrated a similar distribution of NMDA receptors using a single-molecule localization approach (Goncalves, et al., PNAS, 2020). Thus, we did not change their locations in the model.

7. This last point brings another issue: how can we explain that the very first fusion events are distributed throughout the active zone when they are supposed to take place just next to calcium channels?

Response: As mentioned above, the calcium channels were placed arbitrary in the original model, not based on the data. We removed the channels from the model.

However, the reviewer's point here is interesting – why do vesicles fuse synchronously across the active zone in mammalian central synapses. Unlike neuromuscular junctions of frog, fly and worms, calcium channels are not enriched at the center. The freeze-fracture studies seem to indicate that they form clusters but these clusters seem to be distributed throughout the active zone. This organization likely explains the distribution of fusion pits. However, further studies are warranted to reveal the exact locations of calcium channels relative to fusion pits.

Reviewer #2 (Remarks to the Author):

1. The present MS describes experiments that localize AMPA and NMDARs at high resolution in cultured hippocampal synapses. The authors conclude that these distinct types of GluRs have distinct subsynaptic distributions, which has been known for >20 years. The authors also investigated the spatial relationship between these receptors and exocytic pits and concluded that AMPARs are closer than NMDARs to synchronously released vesicles.

Response: We thank the reviewer for the through read of our manuscript and critical feedback.

2. The introduction of the MS is rather biased towards a view that has supporting evidence, but totally ignores a much larger body of work that is inconsistent with this view. One set of published data support the view that the magnitude of the postsynaptic AMPAR-mediated EPSC depends on the nano-arrangement of the location of the vesicle fusion and the clusters of postsynaptic AMPA receptors, whereas there are a large body of experimental and modelling work arguing for a large postsynaptic receptor occupancy and the insignificance of the exact location of the vesicle fusion. The reviewer insists that the Introduction must contain both views, as this issue is far from being concluded.

Response: We apologize for this oversight. For the initial submission, we formatted our manuscript as a short report, instead of an article, and focused on providing the most relevant information to fit the word limit. The reviewer correctly points out that the introduction should be a fair review of the subject. We changed the format of our manuscript and revised the text as follows.

p. 3-4, “A large body of work suggests that the number of these receptors, rather than their location relative to release sites, is the main determinant of how signals are transmitted across synapses. At mammalian central synapses, the number of AMPA and NMDA receptors is highly variable, ranging from zero to several hundreds¹²⁻¹⁸. However, the number scales with the size of postsynaptic densities¹⁶⁻²⁰, and thus, the postsynaptic membrane is highly occupied by the receptors. Given ~2000 glutamate molecules released per fusion event²¹, one would expect these receptors to be activated efficiently regardless of where in the PSD they are located relative to where glutamate is released. In fact, some computer simulations suggest that the receptor number is more important for the amplitude of synaptic signaling than the locations of release^{18,22,23}.

In contrast to this view, recent studies suggest that receptor activation is greatly influenced by receptors forming clusters that are positioned close to release sites²⁴. Due to rapid diffusion^{25,26,27}, the concentration of glutamate necessary for maximal AMPA receptor activation is only achieved at the point of release. In fact, several computer simulations demonstrate that the open probability of AMPA receptors is reduced by 20-70% if located 100 nm away from the point of release^{18,25,28,29}. Several electrophysiology experiments also suggest that glutamate release from single vesicles does not saturate postsynaptic receptors³⁰⁻³². In addition, localization studies suggest that AMPA receptors are clustered within the postsynaptic density^{33,34} and segregated from the NMDA receptor clusters³⁵. Release sites are aligned with clusters of AMPA receptors³⁶, and their association through trans-synaptic adhesion proteins affects the magnitude of synaptic transmission³⁷. Thus, where glutamate is released relative to receptors may be important for their activation.”

In addition, we included a paragraph in discussion to contrast our findings with the literature.

p. 13-14, “The presence of the receptor clusters and their trans-synaptic alignment with release sites suggests that where glutamate is released relative to receptors is likely important for their activation. This is in sharp contrast with the original view that large numbers of receptors are present in the postsynaptic receptive field⁵³ with no particular pattern of localization, and that glutamate released from single vesicles anywhere in the active zone can efficiently activate both AMPA and NMDA receptors^{5,58}. However, several lines of recent data seem to support the idea that release in the proximity of receptors is a major determinant of synaptic strength. First, release evoked by an action potential does not saturate receptor activation³¹. Second, activation of AMPA receptors is sharply dependent on their distances from the point of release^{18,25,28,29}. Third, AMPA receptors need to be in the clusters to enhance the amplitude of signals. Simply increasing the number of AMPA receptors in postsynaptic densities using optogenetics does not change the quantal amplitude nor strength of response from existing synapses, suggesting that the location of receptors is more important⁵⁹. Fourth, the alignment is modulable and can potentially alter synaptic strength^{36,60}. Our data are consistent with these studies and further support the idea that the precise location of release influences efficiency of receptor activation at excitatory synapses. In addition, this is the first demonstration that the NMDA receptor activation is also influenced by such transsynaptic nano-architectures.”

3. The authors also seem to forget a large body of data from the 90's that examined the sub-PSD distribution of AMPA and NMDARs in many brain areas, including the striatum, cortex, hippocampus etc.

Response: We added the following references to the expanded introduction.

12. Nusser, Z., Mulvihill, E., Streit, P. & Somogyi, P. Subsynaptic segregation of metabotropic and ionotropic glutamate receptors as revealed by immunogold localization. *Neuroscience* 61, 421–427 (1994).
13. Popratiloff, A., Weinberg, R. J. & Rustioni, A. AMPA receptor subunits underlying terminals of fine-caliber primary afferent fibers. *J. Neurosci. Off. J. Soc. Neurosci.* 16, 3363–3372 (1996).
14. Kharazia, V. N. & Weinberg, R. J. Tangential synaptic distribution of NMDA and AMPA receptors in rat neocortex. *Neurosci. Lett.* 238, 41–44 (1997).
15. Rubio, M. E. & Wenthold, R. J. Glutamate Receptors Are Selectively Targeted to Postsynaptic Sites in Neurons. *Neuron* 18, 939–950 (1997).
16. Takumi, Y., Ramírez-León, V., Laake, P., Rinvik, E. & Ottersen, O. P. Different modes of expression of AMPA and NMDA receptors in hippocampal synapses. *Nat. Neurosci.* 2, 618–624 (1999).
17. Masugi-Tokita, M. et al. Number and Density of AMPA Receptors in Individual Synapses in the Rat Cerebellum as Revealed by SDS-Digested Freeze-Fracture Replica Labeling. *J. Neurosci.* 27, 2135–2144 (2007).
18. Tarusawa, E. et al. Input-Specific Intrasynaptic Arrangements of Ionotropic Glutamate Receptors and Their Impact on Postsynaptic Responses. *J. Neurosci.* 29, 12896–12908 (2009).

4. The argument of the need of temporal precision for NMDA receptor activation is also biased! The time to first opening distribution of NMDARs is tens of milliseconds. The diffusion/equilibration of glutamate within the cleft is tens of microseconds. The reviewer cannot see why 100 μ s delay in the activation of NMDARs 200 nm away would have any significant effect if the receptors open 20 ms later?

Response: We share the reviewer's concern. It is difficult to imagine how AMPA receptor activation occurring within 100 μ s of glutamate release would impact on the NMDA receptor activation milliseconds later. To address this exact point, we performed the computer simulations (Fig. 3a-b and Supplementary Fig. 4). Our results indicated that the precise timing of the AMPA receptor activation is not important, but the resulting depolarization of the plasma membrane is the key for the NMDA receptor activation. The peak depolarization lasts for 5-8 ms (Supplementary Fig.4h), during which the magnesium block of NMDA receptors is alleviated, thereby increasing the chances of their opening. Since AMPA receptors have a low binding affinity for glutamate, their activation is greatly influenced by where glutamate is released. Based on our simulations, glutamate released 200 nm away from AMPA receptors would reduce the depolarization of the plasma membrane nearly by half, which is consistent with previous studies (Raghavachari and Lisman, 2004, Tarusawa et al., 2009, Uteshev and Pennefather, 1996, Xu-Friedman and Regehr, 2004). Consequently, less NMDA receptors are activated (Fig. 3f). Thus, although not seemingly intuitive, the AMPA receptor activation can increase the NMDA receptor activity when magnesium is present in the extracellular solution.

These points are described in Discussion.

p. 15, "Since the timing of asynchronous release coincides with the peak of postsynaptic membrane depolarization, the magnesium block on NMDA receptors is likely alleviated, allowing them to be activated by asynchronous release. However, it is not the order of release, but rather the activation of the AMPA receptors, that is essential for NMDA receptor activation. Thus, the recently described trans-synaptic alignment of the synchronous release sites with AMPA receptors³⁶ likely ensures the proper activation of AMPA receptors, since their binding affinity for glutamate is very low^{53,54}."

5. The exact proportion of synchronously and asynchronously released vesicles is unknown in this synapse. However, it is well known that most vesicles are release synchronously, unlike in some other synapses (e.g. CCK expressing GABA cells). Thus, the functional consequences of the role of asynchronously released vesicles and their role in NMDA receptor activation remain unknown until this proportion is precisely determined. Why do the authors believe that the arbitrary 5 ms reveals synchronously, whereas 11 ms asynchronously released vesicles?

Response: This is a good point. The temporal assignment for synchronous and asynchronous fusion events should be made clear in the manuscript, instead of simply referencing our recent study, which is now in press at Nature Neuroscience. In short, we separated synchronous and asynchronous events based on the sensitivity to EGTA treatment.

In Kusick et al. (in press at Nature Neuroscience), we characterized synaptic vesicle docking and fusion using the zap-and-freeze device. We stimulated neurons once and froze them 5, 8, 11, and 14 ms later. Due to the mechanics of a high-pressure freezer, the 5 ms time point is

the earliest we can freeze following an action potential induction. Nonetheless, quite a few fusion pits were captured at these time points, with the prevalence peaked at 5 ms and declined to 1/3 of the peak by 11 ms and to the baseline by 14 ms. At 5 ms, roughly 35% of synapses contained one or more fusion pits. This fraction is consistent with the fraction of hippocampal synapses that respond to an action potential, determined using electrophysiology (Allen and Stevens, 1994; Rosenmund, Clements, and Westbrook, 1993; Hessler, Shirke, and Malinow, 1993) and optical methods (Jensen et al., 2019). More importantly, when we performed the same experiments in the presence of EGTA-AM, the number of fusion pits at 5 ms was not affected, suggesting that these are likely the remnants of fusing vesicles during the synchronous phase of release. In contrast, fusion pits were absent at 11 and 14 ms in the presence of EGTA-AM, suggesting that the pits at these time points are likely fusion intermediates from loosely-coupled vesicles (synchronous) or from asynchronous phase of release. However, given the timing of the initiation (5-10 ms after an action potential), we concluded that fusion pits present at 11 ms or later represents asynchronous fusion intermediates.

To make these points clear, we added the following.

p. 9, “Our recent study suggested that fusion intermediates captured at 5 ms represent remnants of synchronously fusing vesicles, while those at 11 ms represent asynchronously fusing vesicles, since the treatment with EGTA-AM only eliminated the latter events⁴.”

Regarding the proportion of synchronous and asynchronous release at these synapses, the number of pits we observed at 11 ms is about 1/3 of the number at 5 ms. However, our assays are not sensitive enough to determine the exact proportion for three reasons. First, we likely missed quite a few pits that had already collapsed by 5 ms. Thus, we cannot determine the total number of fusions. Second, we cannot capture every asynchronous release event since asynchronous release normally lasts for milliseconds to hundreds of milliseconds. Third, we do not know how many synapses had both synchronous and asynchronous release events since we cannot follow the same sets of neurons over time in our assays. Thus, we cannot estimate the proportion based on our assay. However, a study using a similar culture system suggests that postsynaptic currents are reduced by ~20 % when treated with 100 μ M EGTA-AM (Hagler and Goda, 2001), although these may also represent the responses elicited by loosely-coupled vesicles (Grauel, et al., 2016). Thus, a single action potential likely triggers both synchronous and asynchronous release in these synapses, the exact proportion is difficult to determine. To reflect on these points, we amended the text as follows.

p. 26, “An electrical field of 10 V/cm was applied for 1 ms to induce a single action potential, and cells were frozen 5 ms and 11 ms after the stimulus⁴. These time points are chosen based on our recent study suggesting that pits captured at these time points represent fusion intermediates during synchronous and asynchronous release, respectively⁴. The exact proportion of these events could not be determined, but based on EGTA experiments, asynchronous release may account for up to 20% of the currents in these synapses^{72,73}.”

p. 43, “Although shown here as taking place in the same active zone, the degree to which synchronous and asynchronous release may occur at the same active zone after a single action potential is unknown”

6. *Is the probability of finding such pits consistent with the probability of vesicle release?*

Response: As discussed above, ~35% of synapses exhibited pits in our assay. This fraction is consistent with the fraction of hippocampal synapses that respond to an action potential. The three original papers demonstrating synaptic unreliability in hippocampal cells found that synapses typically release neurotransmitter ~30% of the time in response to a single action potential [Hessler, Shirke, and Malinow (1993); Rosenmund, Clements, and Westbrook (1993); Allen and Stevens (1994)]. Experiments using dyes observed a release probability of ~20% [Murthy and Stevens (1997); Ermolyuk ... Volynski (2012); Welzel...Groemer (2012)]. pHluorin assays capable of detecting single vesicle fusions also observed a release probability of ~20% to single action potentials (Leitz and Kavalali 2007). Our assays for the presence of exocytic pits indicate a similar probability for release (~35%) in response to a single action potential. Thus, our observation is quite consistent with the release probability of these synapses. We added the following to the text.

p.26-27, “A 1-ms electrical pulse likely induces an action potential from all neurons on a disk uniformly, and approximately 35 % of synapses exhibit fusion pits⁴ – consistent with the synaptic release probability of these synapses^{74–78}.”

7. *Some of these synapses release vesicles with a probability of <1%! How do the authors handle such synapses?*

Response: The reviewer is correct. Some synapses have an extremely low release probability, and neurons must be stimulated multiple times to induce release from those synapses. Given that we stimulate neurons only once in this study, we are simply detecting events in synapses that have responded – these synapses typically have a high release probability. However, we cannot distinguish the low release probability synapses in our assays.

8. *The authors took a novel approach to label GluA2 containing receptors (His-tag and affinity labelling). They report that 70% of the overexpressed synapses has ~2-3 gold particles. The exact density of AMPA receptors in excitatory synapses is unknown. However, a lower limit can be estimated from the highest labeling achieved so far, that is ~1200 gold/um². If hippocampal synapses are ~0.05-0.1 um², a minimum of 60-120 AMPARs should be present, out of which the authors label 2-3. This low labeling efficacy also affects the outcome of the clustering. Have the authors considered what effect would it have if they analyzed synapses with 100 rather than 10 gold?*

Response: We apologize for the confusion. The average of 2-3 gold particles is per synaptic profile (per image) with a full range of 0-27. From the end-to-end reconstruction of synapses, the average number of AMPA receptors was 16 (median) with a full range of 0-58 – this corresponds to 200 receptors per μm^2 (median, range: 0-1021). These numbers are comparable to those in the published studies using antibodies.

Per synaptic profile

1. Nusser et al., (1994), immuno-EM, ~3 gold particles per synaptic profile with a full range of 0-7, cerebellar parallel fiber synapses.

2. Popratiloff, Weinberg, and Rustioni, (1996), 1.92 and 2.79 gold particles per synaptic profile in C1 and C2 synapses, respectively, of fine-caliber primary afferent fibers.
3. Kharazia and Weinberg (1997), immuno-EM, ~3 gold particles per synaptic profile.
4. Rubio and Wenthold (1997), Immuno-EM, 1.9 gold particles per synaptic profile with a full range of 0-6 in auditory nerve synapses.
5. Takumi et al., (1999) immuno-EM, 2.3 gold particles per synaptic profile with a full range of 0-16.

Per end-to-end synapse

1. Nusser et al., (1998), immuno-EM, 7.6 gold particles per end-to-end reconstruction of synapses (mean) with a full range of 0-87, Schaffer collateral-CA1 synapses – these likely resemble cultured hippocampal synapses. 35.3 gold particles per end-to-end reconstruction of synapses (mean) with a full range of 5-89, adult mossy fiber-CA3 (excluding synapses with less than 5 particles). 22.2 gold particles per end-to-end reconstruction of synapses (mean) with a full range of 7-49, adult mossy fiber-CA3 (excluding synapses with less than 7 particles).
2. Masugi-Tokita et al., (2007), freeze-fracture immuno-gold labelling. 38.1 gold particles per whole postsynaptic density with a full range of 2-178 in cerebellar parallel fiber to Purkinje cell synapses.

Thus, the numbers we observed in our study are comparable to published ultrastructural data.

How do these numbers compare to the estimated number of receptors at synapses? Nusser et al. have estimated that their labeling efficiency is about 43% based on the number (83 receptors per spine) determined from the electrophysiology experiments in hippocampal mossy fiber boutons (Jonas, Major, and Sakmann, 1993). Another electrophysiology-based study by Matsuzaki et al. (2001) also suggests a similar number (82 receptors) in spines of the cultured hippocampal neurons. However, these numbers also include receptors that are outside the postsynaptic density, which are not counted in our study or other ultrastructural studies. The number within the postsynaptic density may be slightly less than ~80. In fact, a single-molecule localization study by Nair et al., (2013) suggested that ~20 receptors are present in each nano-domain (range: 7-40) and 2-3 nanodomains per spine, which would correspond to 40-60 receptors per postsynaptic density on average in cultured hippocampal neurons. In our study, we found 8-10 receptors on average per cluster (range: 4-23) and 2 clusters per end-to-end synapse. Based on these, our estimate for the labeling efficiency in our study is also about 50 %. Given that the distributions of AMPA and NMDA receptors are similar to those revealed by single-molecule localization studies (Goncalves, et al., 2020), we do not expect our conclusions regarding the distributions altered significantly. However, to reflect all these points, we added the following to the text.

p. 8, “The median numbers of AMPA and NMDA receptors were 8 and 6 per cluster (**Fig. 1k**, ranges: 4-23 AMPA receptors and 4-18 NMDA receptors), or 16 and 10 per synapse (**Fig. 1l**, full ranges: 0-58 AMPA receptors and 0-55 NMDA receptors). These numbers correspond to 200 and 135 per μm^2 , respectively (full ranges: 0-1021 AMPA receptors and 0-1079 NMDA receptors) and comparable to previous estimates from the freeze-fracture immuno-gold labelling of adult rat cerebellum⁵⁰ and immuno-electron microscopy of rat hippocampus¹⁹. Since ~20 AMPA receptors are likely available in each nano-domain (ranges: 7.3-42.2)³³, our labeling efficiency is likely ~50 %. Nonetheless, these results are consistent with the single molecule

localization study³⁵ and suggest that AMPA receptors tend to form ~2 clusters (**Fig. 1i**) and surround the NMDA receptor cluster, which is located near the center of the postsynaptic density.”

9. In Figure 1, the authors present one image with which they illustrate that gold particles for AMPARs are located towards the periphery of the synapse. In suppl figure 1 (where they ‘validate’ the specificity of their method) they provide 4 electron micrographs of 4 synapses in which gold particles for AMPARs are rather uniformly distributed; there are just as many golds in the middle than at the edge.

Response: We thank the reviewer for pointing this out. We would like to emphasize that these are *representative* images of synaptic profiles (random 40-nm segments of synapses) from the datasets. These slices are not necessarily going through the center of an active zone/post-synaptic density. Thus, although AMPA receptors are distributed towards the edges of a reconstructed post-synaptic density, gold particles may be found near the center of a post-synaptic density in given images. Given the distribution of AMPA receptors, 34 % the gold particles are expected to be near the center of a postsynaptic density (between the center and the half-way to the edge). In three images shown for the wild-type neurons, there were 12 gold particles in total. Of which, 5 were found in this criterion (42%). We could have cherry-picked images of synaptic profiles that have gold particles accumulated near the edge, but for the sake of reproducibility, we decided to show randomly selected images in the figure. However, the reviewer’s concern is valid. We added a few more images to the Supplementary Figure 1. Now, 9/23 gold particles are found between the center and the half-way to the edge), slightly better matching the distribution described in the manuscript.

10. The identification of the fusion events in the EMs is rather uncertain (Fig 2a, b). When one looks at the presented micrographs, the membranes are totally disintegrated with large, swollen extracellular spaces surrounding the synapses (which are impossible to identify based on the presented images!). In such EM images, the identification of docked vesicles and exocytic pits is problematic, to say the least!

Response: We think that the preservation quality is excellent. Since high-pressure freezing prevents collapse or wash-out of cytoplasmic material, tissues with the best preservation and no ice crystal damage are often of low contrast. The presence of extended extracellular spaces is not due to the poor preservation. In cultures, synapses are sparse, making the extracellular space appear swollen since no other cells are surrounding synapses. However, this is quite normal and would not affect the physiology of neurons. We now uploaded the original images to our figshare site (<https://figshare.com/account/home#/projects/87923>) to illustrate the morphology of samples.

Nonetheless, we share the reviewer’s concern. The pits are very subtle. We described pits as “smooth curvature (not mirrored by the postsynaptic membrane) in an otherwise straight membrane”. By these criteria, membrane curvatures like those illustrated in the Reviewer’s Figure (white arrows) are not considered pits (these images are included in Kusick et al. in press). We note that we could either miss or over-annotate some pits under these criteria, but we tried to

be consistent across all the samples. Importantly, analysis is performed on shuffled images blind to treatment, and the segmentation is always validated by an additional member in the laboratory. Thus, differences between samples cannot be the result of bias or the inability to segment the images accurately due to poor morphology. Moreover, we always included a no-stimulation control, and we consistently observe a lack of pits in the no-stimulation controls and presence of pits in stimulated samples. These are stated in the Methods.

p. 29-30, “Pits were identified as smooth curvature (not mirrored by the postsynaptic membrane) in an otherwise straight membrane. These pits are considered exocytic⁴. Pits outside the active zone are considered endocytic or membrane ruffles, as this is the primary site for ultrafast endocytosis⁵¹. Under these criteria, we could miss or over-annotate vesicles and pits. To minimize the bias and maintain consistency, all image segmentation, still in the form of randomized files, was thoroughly checked by a second member of the lab.”

11. The authors do not provide data about the density of docked vesicles in their AZ, which would allow direct comparisons to published data using chemical fixation or only high pressure freezing. Without knowing that, a median distance of 100 nm from the docked vesicles to the AMPA/NMDARs indicate a maximum of 2-3 docked vesicle per AZ. That is much lower than estimates using the best methods (e.g. Imig et al).

Response: We apologize for not making these numbers readily accessible in the manuscript. The number of docked vesicles per synaptic profile was provided in the Supplementary Figure 2G and Supplementary Table 1 in the previous submission. The average number of docked vesicles per synaptic profile is 1.9, and this number is consistent with all our previous studies (Kusick et al., 2020; Watanabe et al., 2013, 2014, and 2018) as well as those from other labs (Chang, et al., 2018; Nyitrai et al., 2020; Wang et al., 2016). We agree with the reviewer that these numbers are important, and thus, we included the numbers in the Results.

p. 9. “The numbers and distributions of docked vesicles and exocytic pits were all consistent with our previous studies (docked: 1.9 ± 0.05 per synaptic profile; pits: 0.28 ± 0.03 per synaptic profile **Fig. 2c, Supplementary Fig. 3g,h**)^{4,51,52}.”

However, the question is what this number means in terms of the end-to-end reconstruction of synapses. Previous studies suggest that 6-12 vesicles are typically docked and they are uniformly distributed within the active zone (for example, Schikorski and Stevens, 1997; Imig et al. 2014). Likewise, in Kusick et al. (2020), we demonstrated that the median number of docked vesicles per synapse is 10-12. In the current study, since the most data are collected from synaptic profiles and serial-block face data do not have sufficient image resolution to distinguish docked from undocked vesicles, we cannot determine the average number of docked vesicles per synapse. However, given the similarity in the number per synaptic profile among all other studies, we expect 10-12 docked vesicles from reconstructed synapses.

12. The authors interpretation of their own data is interesting: ‘Of note, exocytic pits at 5 ms were distributed throughout the active zone with a slight bias to the center (median = 0.4)’ few

sentences below: ' These results suggest that neurotransmitter is released synchronously near AMPA receptors and asynchronously around NMDA receptors.'

Response: The reviewer correctly points out that the distribution of AMPA receptors within postsynaptic density (towards the edge) does not match the distribution of fusion pits within the active zone during synchronous release (uniform but less at the edge), and thus our interpretation that synchronous release takes place near the AMPA receptors may be incorrect. We agree with the reviewer that this is quite confusing. However, we think that this discrepancy likely comes from the fact that the distributions were measured from single profiles, not from the reconstructed synapses. Single synaptic profiles are *random* representations of synapses; each image represents a 40-nm segment of a synapse, sliced at a *random* location. These slices are not necessarily going through the center of an active zone/post-synaptic density. Thus, although NMDA receptors are localized near the center of a post-synaptic density when reconstructed, they may not appear enriched towards the center in individual images. Likewise, AMPA receptors are distributed towards the edges of a reconstructed post-synaptic density, but gold particles may be found near the center of a post-synaptic density in given images (Supplementary Fig. 1a). Thus, the distributions in single profiles would likely appear different from those determined from the 3D reconstructions.

Nonetheless, the data from single profiles can be used to approximate the distribution when sufficient data are collected. To draw this point clear, we used the datasets from the 3D reconstruction and measured the distribution of receptors to the center of a postsynaptic density in each image, instead of the true center in the reconstructed postsynaptic density (Supplementary Fig. 2a). When we map the first 25 gold particles, the distributions of both AMPA and NMDA receptors are nearly identical (Supplementary Fig. 2a), with their means being both 0.55. This is likely because we are sampling sparsely. With 100 gold particles mapped, the means are 0.54 and 0.44, respectively, which are similar to the means obtained from all particles mapped (0.57 and 0.42). Qualitatively, these data are quite similar to the data obtained from the 3D reconstructions (means: 0.56 and 0.33) and single profile data obtained from random synapses (means: 0.6 and 0.5). Thus, although true distributions of gold particles cannot be determined from single profiles, they can be estimated from large datasets.

The distribution of pits to receptors is determined by calculating the distance from pits to gold particles in each image, rather than calculating the distances from these structures to the pseudo-center of the active zone/postsynaptic density. Thus, regardless of where in active zones images are acquired from, the relative distances of pits to AMPA receptors and NMDA receptors can be determined from the single profiles. Thus, our interpretation regarding the relationship between synchronous fusion events and AMPA receptors is reasonable. To make these points clearer, we amended the text as follows.

p. 9-11, “The numbers and distributions of docked vesicles and exocytic pits were all consistent with our previous studies (docked: 1.9 ± 0.05 per synaptic profile; pits: 0.28 ± 0.03 per synaptic profile **Fig. 2c, Supplementary Fig. 3g,h**)^{4,51,52}. Of note, asynchronous fusion intermediates at 11 ms were strongly biased towards the center (**Fig. 2a-b, d**, median = 0.1, $p < 0.001$, **Supplementary Fig. 3a-d** for more example micrographs). Thus, the distribution of fusion events during asynchronous release is similar to that of NMDA receptors.

To test the spatial relationship between fusion events and receptors, we measured the distance between receptors and docked vesicles or exocytic pits. The median distances from

docked vesicles to AMPA and NMDA receptors were 95 nm and 73 nm, respectively, at rest (**Fig. 2e** inset), and remained largely unchanged following stimulation (**Fig 2e** inset: AMPA receptors, 102 nm at 5 ms; and 92 nm at 11 ms; **Fig. 2e** inset: NMDA receptors, 63 nm at 5 ms, 88 nm at 11 ms). This relationship between docked vesicles and receptors is expected given the uniform distribution of docked vesicles in active zone before and after stimulation (Fig. 2c)⁴.

In contrast, the distribution of exocytic pits relative to receptors was not uniform. During the synchronous phase of release (5 ms time points in our assay), pits were distributed throughout the active zone (**Fig. 2d**). However, when measured relative to each type of receptors, exocytic pits were found closer to AMPA receptors (median = 67 nm) than NMDA receptors (median = 139 nm; **Fig. 2f**). Interestingly, exocytic pits during asynchronous phase of release (11 ms) were distant from AMPA receptors (**Fig. 2f**, median = 120 nm) but closer to NMDA receptors (**Fig. 2f**, median = 56 nm). These results suggest that neurotransmitter is likely released synchronously near AMPA receptors and asynchronously around NMDA receptors.”

Reviewer #3 (Remarks to the Author):

1. This is an interesting paper combining cutting edge EM and very nice Monte Carlo simulations to examine the implications of distinct subsynaptic localization of AMPA and NMDA receptors GluARs and GluNRs). The authors show that synchronous neurotransmitter release occurs in regions of the active zone (AZ) that are directly apposed by GluARs in the postsynaptic density (PSD), whereas GluNRs are localized in distinct clusters that face sites that release asynchronously. The modeling then addresses whether this arrangement permits “priming” of GluNRs (i.e., occupancy of one of the two glutamate binding sites during one release event to facilitate greater activation during the next release event), and also whether it maximizes the capacity of GluARs to depolarize the postsynaptic membrane and relieve the Mg block on GluNRs.

The impact of this study rests upon its ability to overturn about 30 years of thinking in the field that GluARs and GluNRs are effectively co-localized in the synaptic cleft, sufficiently activated by a single vesicle of glutamate, and activated in similar proportions during spontaneous and evoked release (Bekkers and Stevens, 1989, Nature; Bekkers, et al., 1990, PNAS). It also calls into question the even longer-lived tenet, back to Bernard Katz, that evoked responses comprise a slightly asynchronous collection of events accurately represented by those recorded spontaneously as mEPSCs.

Response: We thank the reviewer for appreciating the significance and novelty of our work.

2. The MCell modeling addresses rather exotic ideas at the expense of simpler, more fundamental predictions that can be tested experimentally: First, in Mg-free solutions do mEPSCs exhibit a larger GluNR component (relative to the GluAR component) than evoked EPSCs? Second, do GluNRs see less glutamate than GluARs during an evoked EPSC, and is the opposite the case during mEPSCs? This is a harder question to address experimentally (but see Liu, et al., 1999, Neuron). Given what has come before, the strong expectation is that GluARs and GluNRs are activated to similar relative extents during miniature and evoked EPSCs. If that is, in fact, true, then the anatomical results shown here would appear to reflect a molecular ultrastructure that organizes postsynaptic receptors via distinct mechanisms that nonetheless enables them to be activated similarly in response to spontaneous or evoked release. It is reasonable for the authors to suggest that such electrophysiological studies are beyond the scope of the current study, but their omission diminishes the impact substantially.

Response: We agree with the reviewer that addressing the relationship between spontaneous release sites and receptors would be interesting. As pointed out, one prevailing view that came out of experiments in 80's and 90's was that receptors do not need be precisely aligned with the release sites for activation. This view originated from the data suggesting that 1) enough glutamate molecules are stored in synaptic vesicles (Burger et al. 1989), 2) the postsynaptic receptive field is highly occupied by receptors (Jonas, Major, Sakmann, 1993), 3) spontaneous release from single vesicles can activate both AMPA and NMDA receptors (Bekkers and Stevens, 1989; Bekkers, et al., 1990), and 4) the number of receptors, rather than the location of release, is likely important for the amplitude of postsynaptic currents (Faber et al 1992; Holmes 1995; Wahl, Pouzat, Stratford, 1996).

In contrast to this view, our data now indicate that evoked responses are likely elicited by release from multiple vesicles that fuse either simultaneously (Kusick et al., 2020) or sequentially (Kusick et al., 2020 and this manuscript) and in the latter case, the release sites for synchronous and asynchronous events are aligned with AMPA receptors and NMDA receptors, respectively. If these events shape the postsynaptic currents, how do spontaneous fusions of single vesicles would elicit the response from both types of receptors in the same proportion? This is an interesting question and warrant deep considerations.

The reviewer suggested us to perform two sets of electrophysiology experiments to address this question: 1. Repeat the same experiments as in Bekkers and Stevens, 1989, to determine the relative AMPA and NMDA currents, induced by spontaneous and evoked release in the absence of Mg^{2+} ; 2. Determine the fractions of AMPA and NMDA receptors activated during spontaneous and evoked release. As the reviewer suggests, the first set of experiment is more feasible than the second, but in either case, it is nearly impossible to isolate single synapses and measure the pure responses from the receptors within the postsynaptic density since the local perfusion of Ca^{2+} and glutamate would activate all receptors in spines, even those that are extrasynaptic and potentially at neighboring synapses. In addition, for spontaneous release, we cannot isolate responses from the target synapses even with the local perfusion of Ca^{2+} since some spontaneous release is induced by the fluctuation in the internal calcium source (Sharma & Vijayaraghavan, 2003; Llano et al. 2000). To isolate the response in single synapses, we need to perform simultaneous calcium imaging in spine (Reese and Kavalali, eLife, 2016) or glutamate imaging. Such experiments would require extensive optimizations of the system, and under the current climate, it is difficult to set up those experiments in my lab or find a collaborator as most places are operated at less than 50% capacity.

Thus, instead of electrophysiology, we performed additional modeling experiments to test the AMPA and NMDA receptor contributions during spontaneous release and evoked release. Specifically, in the absence of magnesium, we induced a single vesicle release 1) near AMPA receptors, 2) near NMDA receptors, and 3) at random locations within an active zone, and measured AMPA and NMDA receptor responses in each case. To achieve the same proportion to evoked release (two vesicle release – synchronous and asynchronous), spontaneous release may need to be near the NMDA receptors, since the NMDA/AMPA activation ratio is similar when a single vesicle release is simulated near the NMDA receptors (Supplementary Fig. 4g). These data are now included in the manuscript.

However, accumulating data seem to indicate that spontaneous release uses a distinct pool of synaptic vesicles (Sara et al., 2005; Chung et al., 2010; Fredj and Burrone, 2009), which may not be readily available at active zones, and also activates a distinct set of postsynaptic receptors that are not activated by evoked release (Atasoy, et al. 2010; Sara et al., 2011). Interestingly, the NMDA receptor activation during spontaneous release does not seem to depend on the dendritic depolarization or AMPA receptor activity (Espinosa and Kavalali, 2009). Thus, spontaneous release may elicit the postsynaptic currents elicited by a completely different mechanism. However, these are still contentious and require further testing (Kavalali, 2015). In the future, we would like to characterize the spontaneous release sites, but with our assay, it will require the collection and analysis of over 20,000 images (over 4 times of all images analyzed in this study). Thus, it is beyond the scope of the current study. To reflect all these points, we added a paragraph in Discussion.

p. 11, “Interestingly, the proportion of NMDA and AMPA receptors activated by these two release events was similar to the proportion activated when a single vesicle release occurs near NMDA receptors, but not around AMPA receptors or at random locations (**Supplementary Fig. 4g**). These results suggest that the location of release also influences the activation of NMDA receptors.”

p. 14-15, “Nonetheless, whether glutamate is released spontaneously from single vesicles or actively following an action potential, similar proportions of AMPA and NMDA receptors are thought to be activated⁵. Our data indicate that an action potential may induce glutamate release from two vesicles: synchronously near AMPA receptors and asynchronously near NMDA receptors. Although we do not know how often a synapse releases glutamate both synchronously and asynchronously after a single action potential, evoked release likely leads to greater activation of NMDA receptors, given that synchronous multivesicular release is also quite prominent in these synapses⁴. Therefore, to achieve a similar proportion of activation by single vesicles, glutamate may need to be released at a particular location during spontaneous release. Our computer simulations suggest that a single vesicle release near the NMDA receptors may be able to activate both AMPA and NMDA receptors with a similar proportion to the evoked release, suggesting that spontaneous release may occur near NMDA receptors. However, spontaneous release has been proposed to use a distinct pool of synaptic vesicles⁶¹⁻⁶³, which may not be readily available at active zones. In addition, a distinct set of postsynaptic receptors may be activated by spontaneous release^{64,65}. In fact, the NMDA receptor activated during spontaneous release does not seem to depend on the dendritic depolarization or AMPA receptor activity⁶⁶. Thus, spontaneous release may elicit the postsynaptic currents by a completely different mechanism. However, these ideas are still contentious and require further testing⁶⁷.”

3. p. 20: in the methods, the authors note that the GluAR kinetic parameters used in the simulations were adjusted to fit with recorded mEPSCs. This seems circular. One would presume that this is accomplished by placing the GluARs at some distance from an asynchronous release site and then adjusting the kinetics to match mEPSCs recorded by others. The GluAR kinetic properties were obtained in excised patches (from CA3 pyramidal cells) and so provide good measures of binding and unbinding rates regardless of a particular glutamate waveform. If major adjustments were required, it seems more likely to suggest that the proposed glutamate waveform reaching GluARs during an asynchronous event may be incorrect.

Response: We thank the reviewer for the careful read of the manuscript. This was simply a mistake in the description. We used the published kinetic scheme and kinetic rate constants for AMPAR (GluAR) activation and desensitization by glutamate. We amended the text as follows.

p. 32, “The realistic model of glutamatergic synaptic environment was constructed from 3-D electron microscopy of hippocampal area CA1 neuropil as previously described^{27,80,81}. The kinetic scheme and kinetic rate constants for AMPAR (GluAR) activation and desensitization by glutamate were obtained from previously published reports (see ref⁵³ for details of the kinetic scheme and ref³⁷ for the rate constants). The NMDAR kinetics were obtained from Vargas-Caballero and Robinson³⁹.”

Minor comment:

4. p. 11: the point about GluAR binding affinity being low is potentially misleading here, as affinity is measured at equilibrium, and a synaptic event is far too brief to approach equilibrium. In the 100 us of a synaptic event, the activation of postsynaptic receptors depends primarily on their glutamate binding rate, which is actually higher for GluARs than for GluNRs.

Response: The reviewer is correct. In fact, the binding kinetics used in this study are similar for both types of receptors (1.5 and 1.1×10^7). We removed the point about AMPA receptor binding affinity from the sentence.

p. 15, “Thus, the recently described trans-synaptic alignment of the synchronous release sites with AMPA receptors¹⁴ likely ensures the proper activation of AMPA receptors, ~~since their binding affinity for glutamate is very low~~^{32,33,}”

Reviewers' Comments:

Reviewer #1:

Remarks to the Author:

The authors have addressed all my points satisfactorily.

Benoît Zuber

Reviewer #2:

Remarks to the Author:

The reviewer acknowledges the changes to the MS regarding a more balanced view on the literature.

'Thus, although not seemingly intuitive, the AMPA receptor activation can increase the NMDA receptor activity when magnesium is present in the extracellular solution.'

This is a valid argument. The reviewer agrees with this. However, this has nothing to do with the issue whether NMDARs should have any nonrandom arrangement in the PSD or not.

6. Is the probability of finding such pits consistent with the probability of vesicle release?

'Response: As discussed above, ~35% of synapses exhibited pits in our assay. This fraction is consistent with the fraction of hippocampal synapses that respond to an action potential. The three original papers demonstrating synaptic unreliability in hippocampal cells found that synapses typically release neurotransmitter ~30% of the time in response to a single action potential [Hessler, Shirke, and Malinow (1993); Rosenmund, Clements, and Westbrook (1993); Allen and Stevens (1994)]. Experiments using dyes observed a release probability of ~20% [Murthy and Stevens (1997); Ermolyuk ... Volynski (2012); Welzel...Groemer (2012)]. pHluorin assays capable of detecting single vesicle fusions also observed a release probability of ~20% to single action potentials (Leitz and Kavalali 2007). Our assays for the presence of exocytic pits indicate a similar probability for release (~35%) in response to a single action potential. Thus, our observation is quite consistent with the release probability of these synapses. We added the following to the text.'

'First, we likely missed quite a few pits that had already collapsed by 5 ms. Thus, we cannot determine the total number of fusions.'

Based on the above argument, it seems that the authors do not miss events in the first 5 ms. If a significant portion of the released vesicles had disappeared by the time of the fastest freezing (5 ms), then the Pr must have been much larger.

8th point: The reviewer is now convinced that the labeling efficiency of the present study is not very low, based on the argument of the authors and the correction of the text.

9. The reviewer agree that cherry picking is not the correct way! This is not what she/he suggested. The point is that if randomly selected images apparently show the opposite of the main conclusion, that certainly needs some explanation.

10. The authors argue that the 'preservation quality is excellent'. The reviewer has a different view on this; c.f. Fig 2a top row middle panel and bottom row right panel to Watanabe et al., 2013 Nature fig 1c-e and fig 2a (and to several other relevant publications).

12. The authors argument regarding the interpretation of their data is circumstantial. Their argument might explain the data and consistent with their main conclusion, but one could come up with many other explanations that do not support the main conclusion.

Reviewer #3:

Remarks to the Author:

The authors have responded to some of my previous concerns but have declined to perform electrophysiological experiments to confirm the model's predictions. I have a great deal of regard for the previous work by the authors, both in the EM and in Monte Carlo modeling of synaptic transmission. It is, therefore, with considerable respect that I nonetheless remain unconvinced that the anatomical arrangement demonstrated here translates to the physiological effects predicted by the model.

Previous work has argued that the relative location within the synapse of the release site and AMPA receptor clusters influences strongly the amplitude of the AMPA receptor postsynaptic conductance. Moreover, previous work also has suggested that synchronous and asynchronous/spontaneous release activates different receptor populations within the synapse. This paper supports these previous conclusions, but the novelty here lies in the argument that synchronous release preferentially activates AMPA receptors so that they can depolarize the postsynaptic membrane sufficiently to relieve Mg²⁺ block of NMDA receptors located in the same synapse. For this to work, the postsynaptic AMPA receptors need to be activated sufficiently (i.e., by a vesicle released directly overhead) to depolarize the postsynaptic membrane enough to relieve Mg²⁺ block of NMDA receptors. Accordingly, the simulations suggest that glutamate released from a single vesicle depolarizes the postsynaptic membrane by 30 mV (Supplementary Figure 4h). According to Jahr and Stevens (1990), in physiological Mg²⁺ (1 mM) a depolarization from -70 to -40 mV would increase the NMDA receptor conductance from 4.5% to 23% of maximum, a five-fold increase.

I am unaware of experimental evidence in the literature that release of a single vesicle can evoke such a large depolarization. On the contrary, voltage recordings from the dendrites of hippocampal CA1 pyramidal cells indicate that unitary (likely single-vesicle) EPSPs are 30-150 times smaller (Magee and Cook, 2000, *Nature Neuroscience*). In that paper, single-vesicle synaptic events, recorded in the soma, were about 0.2 mV in amplitude, consistent with previous recordings (e.g., Sayer, et al., 1990, *J. Neurosci.*). EPSPs recorded in the dendrites were larger, as expected, and varied roughly four-fold with dendritic location, but at their largest they were only 0.8 mV. Such a small depolarization would increase the NMDA receptor conductance almost negligibly (to 4.66%). These dendritically recorded EPSPs were about as fast as those simulated here, suggesting that they were not likely to have been attenuated significantly over the short distance between the synapse and the dendritic electrode.

I apologize to the authors for not raising this particular point in my previous review, but these considerations, together with my previous concerns regarding discrepancies between the simulations and previous electrophysiological work, casts substantial doubts on the modeling results presented here. I have no reason to quibble with the anatomical results, but I remain unconvinced of their physiological significance.

Point-by-point response to reviewers

Nature communication manuscript: NCOMMS-20-19533-T

Reviewers' comments *italicized*.

Reviewers' comments have been renumbered for easy reference.

Responses are labeled.

Reviewer #1 (Remarks to the Authors)

The authors have addressed all my points satisfactorily

Response: We thank Dr. Zuber for the positive outlook of our manuscript

Reviewer #2 (Remarks to the Authors)

1. The reviewer acknowledges the changes to the MS regarding a more balanced view on the literature

Response: We thank the reviewer for the thorough read of our manuscript

2. 4th point: the question was “The argument of the need of temporal precision for NMDA receptor activation is also biased! The time to first opening distribution of NMDARs is tens of milliseconds. The diffusion/equilibration of glutamate within the cleft is tens of microseconds. The reviewer cannot see why 100 us delay in the activation of NMDARs 200 nm away would have any significant effect if the receptors open 20 ms later?” Our response was ‘Since AMPA receptors have a low binding affinity for glutamate, their activation is greatly influenced by where glutamate is released. Based on our simulations, glutamate released 200 nm away from AMPA receptors would reduce the depolarization of the plasma membrane nearly by half, which is consistent with previous studies (Raghavachari and Lisman, 2004, Tarusawa et al., 2009, Uteshev and Pennefather, 1996, Xu-Friedman and Regehr, 2004). Consequently, less NMDA receptors are activated (Fig. 3f). Thus, although not seemingly intuitive, the AMPA receptor activation can increase the NMDA receptor activity when magnesium is present in the extracellular solution.’

A new comment from the reviewer: This is a valid argument. The reviewer agrees with this. However, this has nothing to do with the issue whether NMDARs should have any nonrandom arrangement in the PSD or not.

Response: The reviewer is correct. The described mechanism can activate NMDA receptors that are found anywhere within the postsynaptic density, and does not explain this central

organization and trans-synaptic alignment with asynchronous release. To further the understanding, we performed an additional modeling experiment to test the effect of two consecutive release outside the NMDA receptor clusters, near the AMPA receptor clusters (Fig. 3F). This release pattern activated NMDA receptors to a greater degree following a single stimulus, presumably because membranes are further depolarized due to the activation of more AMPA receptors. However, this release pattern seems to increase the number of desensitized AMPA receptors from a single stimulus, and therefore, the NMDA receptor activation would be attenuated if more stimuli are applied. Thus, it is quite possible that this spatial and temporal pattern of release we discovered here balances the activation of AMPA and NMDA receptors. To reflect on these points, we expanded our discussion to include the following.

p. 14-15, “NMDA receptors have a higher affinity for glutamate, and thus their locations within a postsynaptic density are thought to be less critical. Previous simulations demonstrated that whether they are in the cluster or randomly localized, NMDA receptors can be activated equally well from a single vesicle release³⁵. Our data here also suggest that locations of release are less important for NMDA receptor activation as long as AMPA receptors are activated. In fact, NMDA receptors are activated to a greater degree when both synchronous and asynchronous release occurs near AMPA receptors. However, this release pattern would increase the number of desensitized AMPA receptors, leading to faster depression at these synapses. Thus, one release event near AMPA receptors and another release event near NMDA receptors likely maximize the membrane depolarization and NMDA receptor activation, while ensuring that a sufficient number of naïve AMPA receptors are available to respond to the next stimulus.”

However, we agree with the reviewer that the activation of NMDA receptors may only be one role of this trans-synaptic organization. We would like to emphasize that the current study reports a novel trans-synaptic organization of release sites and receptors and only explores one potential aspect of this organization in synaptic transmission but does not exclude possibilities for other roles. To explore other possibilities, we will need to set up experiments that specifically manipulate asynchronous release, NMDA receptor organization, or trans-synaptic alignments. We are currently investigating proteins essential for each of these. These experiments are not central to the current manuscript but will be required to further the understanding of the trans-synaptic organization of release sites and receptors in synaptic transmission. To reflect on these points, we added the following to Discussion.

p. 15-16, “Here, we propose that trans-synaptic organization of NMDA receptors may allow them to be activated during asynchronous release. However, it is possible that this organization may serve different functions. To test this point, the actual functional output from NMDA receptors must be measured experimentally while manipulating their organization. Currently, it is unknown how asynchronous release sites are aligned with NMDA receptors. One possibility is that this organization arises by coincidence, that is, asynchronous release sites simply occur

further away from synchronous release sites, which may be aligned with AMPA receptor clusters via neuroligin-1³⁷, and coincide with NMDA receptors. Alternatively, many synaptic adhesion molecules exist, and they all interact with the presynaptic release machinery as well as postsynaptic receptors and their scaffolding proteins^{37,24,68,69}. Thus, it is tempting to speculate that the arrangement of these molecules give rise to this unique trans-synaptic organization of release sites and receptors at excitatory synapses.”

3. 6th point.. *Is the probability of finding such pits consistent with the probability of vesicle release?*

'Response: As discussed above, ~35% of synapses exhibited pits in our assay. This fraction is consistent with the fraction of hippocampal synapses that respond to an action potential. The three original papers demonstrating synaptic unreliability in hippocampal cells found that synapses typically release neurotransmitter ~30% of the time in response to a single action potential [Hessler, Shirke, and Malinow (1993); Rosenmund, Clements, and Westbrook (1993); Allen and Stevens (1994)]. Experiments using dyes observed a release probability of ~20% [Murthy and Stevens (1997); Ermolyuk ... Volynski (2012); Welzel...Groemer (2012)]. pHluorin assays capable of detecting single vesicle fusions also observed a release probability of ~20% to single action potentials (Leitz and Kavalali 2007). Our assays for the presence of exocytic pits indicate a similar probability for release (~35%) in response to a single action potential. Thus, our observation is quite consistent with the release probability of these synapses. We added the following to the text.'

'First, we likely missed quite a few pits that had already collapsed by 5 ms. Thus, we cannot determine the total number of fusions.'

Based on the above argument, it seems that the authors do not miss events in the first 5 ms. If a significant portion of the released vesicles had disappeared by the time of the fastest freezing (5 ms), then the Pr must have been much larger.

Response: The reviewer is correct. Based on the known Pr of these synapses, it is quite likely that this time point captures at least one fusion event in *all* synapses that have responded to an action potential. However, we cannot be sure that *all* fusions are captured in those responded synapses (i.e. we may have missed multivesicular events). Based on loss of docking, we expected to see about 3-4 fusion events per synapse, but we only captured 1-2 fusion pits (Kusick et al., Nature Neuroscience 2020). In Kusick et al., we attributed some of these missing vesicles to activity-dependent undocking events, but it is equally likely that some had undergone fusion and we missed those events. Thus, we cannot claim we capture all fusion events.

4. 8th point: *The reviewer is now convinced that the labeling efficiency of the present study is not very low, based on the argument of the authors and the correction of the text*

Response: We thank the reviewer for acknowledging and accepting our changes.

5. 9th point: *The reviewer agree that cherry picking is not the correct way! This is not what she/he suggested. The point is that if randomly selected images apparently show the opposite of the main conclusion, that certainly needs some explanation.*

Response: We thank the reviewer for pointing this out! As noted in the last round of rebuttal, we made sure the images do represent the data.

6. 10th point: *The authors argue that the ‘preservation quality is excellent’. The reviewer has a different view on this; c.f. Fig 2a top row middle panel and bottom row right panel to Watanabe et al., 2013 Nature fig 1c-e and fig 2a (and to several other relevant publications).*

Response: We apologize for the overstatement in the rebuttal. We agree with the reviewer that the quality of some images is not ideal. For those images mentioned, we went back and looked for more images, but they are either similar in quality or not representative. Thus, we will keep those images. We would like to emphasize, however, that the images from single experiments are pooled, randomized, and analyzed blind, and experiments were repeated several times. Thus, we are confident about our data.

7. 12th point. *The original question/comment, “The authors interpretation of their own data is interesting: ‘Of note, exocytic pits at 5 ms were distributed throughout the active zone with a slight bias to the center (median = 0.4)’ few sentences below: ‘These results suggest that neurotransmitter is released synchronously near AMPA receptors and asynchronously around NMDA receptors.’ The authors argument regarding the interpretation of their data is circumstantial.*

The new comment: Their argument might explain the data and consistent with their main conclusion, but one could come up with many other explanations that do not support the main conclusion.

Response: We thank the reviewer for acknowledging that our interpretation can potentially explain the data and support the main conclusion. We are excited about our recent finding that synchronous and asynchronous release sites are segregated within the active zone (Kusick et al., Nature Neuroscience, 2020). In this manuscript, we are adding another dimension to this finding – AMPA and NMDA receptors are also segregated within the postsynaptic density - consistent with the recently published data (Goncalves, PNAS, 2020). In the light of another paper, demonstrating the trans-synaptic alignment of synchronous release sites and AMPA receptors (Tang, et al., 2016), we tested whether these receptors are aligned to release sites. Our data additionally demonstrate that fusion pits at 5 ms (EGTA-insensitive release) and 11 ms (EGTA-sensitive release) are found near AMPA and NMDA receptors, respectively. We take these data to suggest that synchronous release aligns with AMPA receptors, and asynchronous release takes

place near NMDA receptors. As the reviewer suggests, there are many interpretations to the data, but we are relieved that the reviewer finds this logic and argument reasonable.

We imagine that we will need to visualize both AMPA and NMDA receptors at the same time and test how close these receptors are to synchronous and asynchronous release sites. Ultimately, we will also need to develop a functional assay to test whether this nanoscopic organization of release sites and receptors has an impact on synaptic transmission. We are psyched about setting up these experiments in our lab and performing those experiments in the future. At the same time, we are also happy about the possibilities that other labs would be able to access these data, think about our models, and come up with clever experiments to test our models. Given that these functional experiments are beyond the scope of the current study, we toned down our conclusions regarding the role of trans-synaptic alignment in synaptic transmission being just for the activation of NMDA receptors. We amended the title and texts as follows.

Title: “**Asynchronous release sites align with NMDA receptors**”

p. 13, “Together, these results suggest that the trans-synaptic alignment of release sites and receptors **likely** ensures the maximal depolarization through AMPA receptors and thereby efficient activation of NMDA receptors, while avoiding saturation of AMPA receptors from a single stimulus.”

p. 2, Abstract, “Computational simulations indicate that this spatial and temporal arrangement of release ~~ensures~~ **can lead to** maximal membrane depolarization through AMPA receptors, alleviating the pore-blocking magnesium leading to greater activation of NMDA receptors. Together, these results suggest that release sites are **likely** organized to activate NMDA receptors efficiently.”

p.5, “Computer simulations suggest that this organization **can induce** membrane depolarization through the AMPA receptors and activate NMDA receptors more efficiently. These data indicate that **one potential role of this spatial organization** of synchronous and asynchronous release sites is to prime NMDA receptors for activation.”

p.15-16, “Here, we **propose that** trans-synaptic organization of NMDA receptors may allow them to be activated during asynchronous release. However, **it is possible that this organization may serve different functions**. To test this point, the actual functional output from NMDA receptors must be measured experimentally while manipulating their organization. Currently, it is unknown how asynchronous release sites are aligned with NMDA receptors. One possibility is that this organization arises by coincidence, that is, asynchronous release sites simply occur further away from synchronous release sites, which may be aligned with AMPA receptor clusters via neuroligin-1³⁷, and coincide with NMDA receptors. Alternatively, many synaptic adhesion

molecules exist, and they all interact with the presynaptic release machinery as well as postsynaptic receptors and their scaffolding proteins^{37,24,68,69}. Thus, it is **tempting to speculate** that the arrangement of these molecules give rise to this unique trans-synaptic organization of release sites and receptors at excitatory synapses.”

We thank the reviewer for the helpful comments and their insights.

Reviewer #3 (Remarks to the Author):

1. The authors have responded to some of my previous concerns but have declined to perform electrophysiological experiments to confirm the model's predictions. I have a great deal of regard for the previous work by the authors, both in the EM and in Monte Carlo modeling of synaptic transmission. It is, therefore, with considerable respect that I nonetheless remain unconvinced that the anatomical arrangement demonstrated here translates to the physiological effects predicted by the model.

Previous work has argued that the relative location within the synapse of the release site and AMPA receptor clusters influences strongly the amplitude of the AMPA receptor postsynaptic conductance. Moreover, previous work also has suggested that synchronous and asynchronous/spontaneous release activates different receptor populations within the synapse. This paper supports these previous conclusions, but the novelty here lies in the argument that synchronous release preferentially activates AMPA receptors so that they can depolarize the postsynaptic membrane sufficiently to relieve Mg²⁺ block of NMDA receptors located in the same synapse. For this to work, the postsynaptic AMPA receptors need to be activated sufficiently (i.e., by a vesicle released directly overhead) to depolarize the postsynaptic membrane enough to relieve Mg²⁺ block of NMDA receptors. Accordingly, the simulations suggest that glutamate released from a single vesicle depolarizes the postsynaptic membrane by 30 mV (Supplementary Figure 4h). According to Jahr and Stevens (1990), in physiological Mg²⁺ (1 mM) a depolarization from -70 to -40 mV would increase the NMDA receptor conductance from 4.5% to 23% of maximum, a five-fold increase. I am unaware of experimental evidence in the literature that release of a single vesicle can evoke such a large depolarization. On the contrary, voltage recordings from the dendrites of hippocampal CA1 pyramidal cells indicate that unitary (likely single-vesicle) EPSPs are 30-150 times smaller (Magee and Cook, 2000, Nature Neuroscience). In that paper, single-vesicle synaptic events, recorded in the soma, were about 0.2 mV in amplitude, consistent with previous recordings (e.g., Sayer, et al., 1990, J. Neurosci.). EPSPs recorded in the dendrites were larger, as expected, and varied roughly four-fold with dendritic location, but at their largest they were only 0.8 mV. Such a small depolarization would increase the NMDA receptor conductance almost negligibly (to 4.66%). These dendritically recorded EPSPs were about as fast as those simulated here, suggesting that they were not likely to have been attenuated significantly over the short distance between the synapse and the

*dendritic
electrode.*

I apologize to the authors for not raising this particular point in my previous review, but these considerations, together with my previous concerns regarding discrepancies between the simulations and previous electrophysiological work, casts substantial doubts on the modeling results presented here. I have no reason to quibble with the anatomical results, but I remain unconvinced of their physiological significance.

Response: The reviewer is correct. The heroic dendritic patch-clamping experiments by Magee and Cook (2001) suggest that the EPSP amplitude measured at dendrites is indeed low (about 0.25- 0.8 mV). However, a subsequent paper by the same author (Magee) and his colleagues demonstrates that the voltage change induced by a unitary synaptic input is ~50x higher at spine heads than at dendrites (Harnett, et al., Nature, 2012). In this paper, they performed two sets of measurements. First, they measured the level of calcium signaling at dendritic spines using calcium dye (Oregon Green BAPTA-1), while stimulating a unitary synaptic input with glutamate uncaging. To isolate the calcium signaling due to the direct depolarization of the spine membrane by synaptic input, they blocked calcium influx through NMDA receptors with AP-5 and back-propagating action potentials using TTX. As in the previous paper, dual patch-recordings of dendrites were used to monitor the voltage change in dendrites. This measurement allowed them to determine the degree of calcium signaling through AMPA-mediated depolarization in spines. Second, they applied an EPSC-shaped waveform from one pipet and measured the resulting voltage change with another pipet. The magnitude of the current injection was set so that the degree of calcium signaling in each spine was similar to that induced by a unitary synaptic input. Given that there is no voltage attenuation going from dendrites to spines (Rall, 1962; Koch and Zador, 1993), the voltage measured at dendrites would approximate the voltage change in spine heads in this case. With this set of measurements along with computer simulations, they demonstrated that the spine must be depolarized by ~25 mV to achieve the degree of calcium signaling induced by an average unitary synaptic input. This number is consistent with the voltage change observed in spines of cortical pyramidal neurons using a voltage-sensitive dye (up to 20 mV; Palmer and Stuart, The Journal of Neuroscience, 2009), suggesting that the change in the membrane potential in spines is likely quite high. We would like to note that our simulations to generate the EPSP waveform in Supplementary Fig. 4h is in line with the simulations of Harnett et al. (2012). We have now referenced these papers in Methods and amended the text as follows.

p. 33, “The membrane potential waveform simulated following a single vesicle release is consistent with previous estimates of voltage change in spines^{83,84}.”

However, we do agree with the reviewer that some functional outputs would strengthen the conclusion of our paper. For this, we can imagine performing electrophysiology experiments to

measure NMDA currents from single synapses in the presence of Mg^{2+} , while blocking asynchronous release. However, this experiment is not feasible at this point for three reasons. First, NMDA currents from single synapses in the presence of Mg^{2+} are too small to be detected by the whole-cell recordings. Second, the use of EGTA-AM can block asynchronous release from presynapses but also directly attenuates the NMDA currents. Third, calcium sensors for asynchronous release are still hotly debated, and thus selective block of asynchronous release through genetic manipulations is also not feasible – in fact, we have a 5-year grant, specifically aiming to address this issue. For these reasons, we decided to perform computer simulations in the first place. It is our hope that the data presented here will stimulate development of new approaches and further the understanding in this area.

Given that these functional experiments cannot be readily performed, we toned down our conclusions regarding the role of trans-synaptic alignment in synaptic transmission being just for the activation of NMDA receptors. We amended the title and texts as follows.

Title: “**Asynchronous release sites align with NMDA receptors**”

p. 13, “Together, these results suggest that the trans-synaptic alignment of release sites and receptors **likely** ensures the maximal depolarization through AMPA receptors and thereby efficient activation of NMDA receptors, while avoiding saturation of AMPA receptors from a single stimulus.”

p. 2, Abstract, “Computational simulations indicate that this spatial and temporal arrangement of release ~~ensures~~ **can lead to** maximal membrane depolarization through AMPA receptors, alleviating the pore-blocking magnesium leading to greater activation of NMDA receptors. Together, these results suggest that release sites are **likely** organized to activate NMDA receptors efficiently.”

p.5, “Computer simulations suggest that this organization **can induce** membrane depolarization through the AMPA receptors and activate NMDA receptors more efficiently. These data indicate that **one potential role of this spatial organization** of synchronous and asynchronous release sites is to prime NMDA receptors for activation.”

p.15-16, “Here, we propose that trans-synaptic organization of NMDA receptors may allow them to be activated during asynchronous release. However, it is possible that this organization may serve different functions. To test this point, the actual functional output from NMDA receptors must be measured experimentally while manipulating their organization. Currently, it is unknown how asynchronous release sites are aligned with NMDA receptors. One possibility is

that this organization arises by coincidence, that is, asynchronous release sites simply occur further away from synchronous release sites, which may be aligned with AMPA receptor clusters via neuroligin-1³⁷, and coincide with NMDA receptors. Alternatively, many synaptic adhesion molecules exist, and they all interact with the presynaptic release machinery as well as postsynaptic receptors and their scaffolding proteins^{37,24,68,69}. Thus, it is tempting to speculate that the arrangement of these molecules give rise to this unique trans-synaptic organization of release sites and receptors at excitatory synapses.”

Reviewers' Comments:

Reviewer #2:

Remarks to the Author:

The reviewer accepts most arguments of the authors.

Reviewer #3:

Remarks to the Author:

The authors have addressed my previous concerns.

Point-by-point response to the reviewer's comments

Reviewer #2 (Remarks to the Author):

The reviewer accepts most arguments of the authors.

Response: We thank reviewer #2 for the through read of our manuscript and improving our manuscript.

Reviewer #3 (Remarks to the Author):

The authors have addressed my previous concerns.

Response: We thank the reviewer for accepting our rebuttal and providing many insights on the topic.